# Modeling microcephaly with cerebral organoids reveals a WDR62–CEP170–KIF2A pathway promoting cilium disassembly in neural progenitors

Wei Zhang[1,10], Si-Lu Yang[2,10], Mei Yang[1], Stephanie Herrlinger[2], Qiang Shao [1], John L. Collar [2], Edgar Fierro[2], Yanhong Shi[3], Aimin Liu [4], Hui Lu[5], Bruce E. Herring [6], Ming-Lei Guo[7], Shilpa Buch[7], Zhen Zhao[8], Jian Xu[1], Zhipeng Lu[9] & Jian-Fu Chen[1]

Primary microcephaly is caused by mutations in genes encoding centrosomal proteins including WDR62 and KIF2A. However, mechanisms underlying human microcephaly remain elusive. By creating mutant mice and human cerebral organoids, here we found that *WDR62* deletion resulted in a reduction in the size of mouse brains and organoids due to the disruption of neural progenitor cells (NPCs), including outer radial glia (oRG). WDR62 ablation led to retarded cilium disassembly, long cilium, and delayed cell cycle progression leading to decreased proliferation and premature differentiation of NPCs. Mechanistically, WDR62 interacts with and promotes CEP170's localization to the basal body of primary cilium, where CEP170 recruits microtubule-depolymerizing factor KIF2A to disassemble cilium. WDR62 depletion reduced KIF2A's basal body localization, and enhanced KIF2A expression partially rescued deficits in cilium length and NPC proliferation. Thus, modeling microcephaly with cerebral organoids and mice reveals a WDR62-CEP170-KIF2A pathway promoting cilium disassembly, disruption of which contributes to microcephaly.

[1] Center for Craniofacial Molecular Biology, University of Southern California (USC), Los Angeles, CA 90033, USA. [2] Department of Genetics, University of Georgia, Athens, GA 30602, USA. [3] Division of Stem Cell Biology Research, Department of Developmental and Stem Cell Biology, Beckman Research Institute of City of Hope, Duarte, CA 91010, USA. [4] Department of Biology, Eberly College of Science, The Pennsylvania State University, University Park, PA 16802, USA. [5] Department of Pharmacology and Physiology, The George Washington University, Washington, DC 20037, USA. [6] Department of Biological Sciences, University of Southern California, Los Angeles, CA 90089, USA. [7] Department of Pharmacology and Experimental Neuroscience, University of Nebraska Medical Center, Omaha, NE 68198, USA. [8] Zilkha Neurogenetic Institute, Keck School of Medicine, University of Southern California, Los Angeles, CA 90033, USA. [9] Department of Pharmacology and Pharmaceutical Sciences, University of Southern California, Los Angeles, CA 90033, USA. [10] These authors contributed equally: Wei Zhang, Si-Lu Yang. Correspondence and requests for materials should be addressed to J.-F.C. (email: Jianfu@usc.edu)

Autosomal recessive primary microcephaly (MCPH) is a congenital brain disorder characterized by a reduction in brain size without severe disruption of brain structures[1–3]. MCPH is considered to be caused by the disruption of neural progenitor cells (NPCs). In the developing cerebral cortex, there are different types of NPCs, named radial glia cells (RGs). The cell bodies of the apical RGs (aRGs) reside in the apical side of the ventricular zone (VZ) of the developing brain. The aRGs produce neurons directly or indirectly via the intermediate progenitor cells (IPCs) residing in the subventricular zone (SVZ)[4]. Recent studies uncovered a primate-enriched neurogenic area, named the outer subventricular zone (oSVZ), which is basal to the classic SVZ. The oSVZ neural progenitors contain transient amplifying IPCs and outer radial glia cells (oRGs), and have been proposed to contribute to the majority of upper-layer neurons[5–7]. Whereas IPCs are conserved between human and mice, oRGs abundantly occur in human cerebral cortex with limited presence in rodent cortex[8], which may potentially explain why MCPH gene mutations cause milder microcephaly phenotypes in mice than that in humans. However, how MCPH mutations affect different types of human NPCs leading to reduced brain sizes remain poorly understood.

Interestingly, most MCPH proteins localize to the centrosomes or spindle poles[1,3]. A variety of functions have been reported for MCPH proteins, including the regulation of centriole duplication, centriole engagement and cohesion, centrosome maturation, centrosome asymmetry, spindle assembly, and astral microtubule dynamics, among others[1,9]. Disruption of these centrosome/spindle-related functions are linked to the premature switch from symmetric to asymmetric cell division, mitotic entry defects, mitotic delay and arrest, or apoptotic cell death, resulting in NPC disruption and smaller brain size[10–14]. It remains unclear how these diverse functions of MCPH proteins observed in rodent models are manifested and disrupted in human NPCs, leading to microcephaly. To address these questions and bridge the gap between mouse models and human diseases, cerebral organoids generated from pluripotent stem cells (PSCs) are emerging as a promising approach to investigate disease mechanisms in a relevant cellular and genetic context[15–18].

The primary cilium is a microtubule-based organelle that functions as a cellular antenna, sensing and transducing different signals during development and tissue homeostasis[2,19–21]. Ciliogenesis is comprised of two phases: cilium assembly and cilium disassembly. During the cell cycle exit in the G0 quiescent phase, the centrosome migrates to the apical cell surface, where the mother centriole is transformed into the basal body to initiate cilium assembly. Upon cell cycle re-entry, the cilium is disassembled, followed by the G1/S phase transition for cell cycle progression[22]. It has been suggested that the primary cilium functions as a checkpoint or brake for cell cycle re-entry. A delay or failure of cilium disassembly would prolong the G0/G1 phase and slow down cell cycle re-entry and progression[23–25]. MCPH-associated proteins have been reported to regulate primary cilium biogenesis coupled with cell proliferation[23,25,26]. In contrast to the diverse functions of MCPH proteins in centrosome biology, their potential roles in cilium and microcephaly are relatively less studied.

Mutations in WDR62 (MCPH2; OMIM 604317) are the second most common genetic cause of MCPH in humans[27–29]. Mouse genetic studies suggested that Wdr62 deletion reduces NPCs and leads to a smaller brain size[12–14]. Wdr62 mutant mice exhibit a mild microcephaly phenotype, suggesting that certain aspects of human WDR62 biology may not be adequately modeled in mice. Wdr62 regulates spindle assembly, spindle orientation, centriole duplication, asymmetric centrosome inheritance, and maintenance, as well as glial cell growth[13,14,30–33]. However, whether Wdr62 functions in the primary cilium remains unknown. To model human microcephaly, we developed cerebral organoids from WDR62-deficient human pluripotent stem cells (hPSCs). Organoid and mouse genetic studies showed that WDR62 depletion resulted in smaller cerebral organoid and brain sizes due to the disruption of NPCs, including the reduced proliferation of oRGs. Mechanistic studies revealed that WDR62 promotes centrosome protein CEP170's localization to the basal body of the primary cilium, where CEP170 recruits KIF2A, a microtubule depolymerizing factor, to disassemble the cilium. We identified a WDR62–CEP170–KIF2A microcephaly protein pathway promoting cilium disassembly, disruption of which contributes to microcephaly.

## Results

**WDR62 mutant cerebral organoids model human microcephaly.** To delete the human WDR62 gene, we generated mutant hPSC cell lines using a clustered regularly interspaced short palindromic repeats (CRISPR)/Cas9 approach[34]. The editing efficiency of gRNA was validated using a T7 Endonuclease I assay. We generated three independent hPSC clones, which were derived from induced pluripotent stem cells (iPSCs) or human embryonic stem (hES) cells. WDR62 mutations occurred as an 8 bp deletion in exon 1 or a 10 or 19 bp deletion in exon 11 (Fig. 1a), all of which resulted in a frameshift and led to premature stop codon generation. Western blot confirmed the absence of WDR62 protein in mutant human PSCs (Fig. 1b). Consistent with its identity as a centrosome protein, WDR62 localized to the centrosome or spindle poles at different phases of the cell cycle in human NPCs (Supplementary Fig. 1). In addition, WDR62 was also detected in the basal body of the primary cilium in wild type but not mutant human NPCs (Fig. 1c), which further validated its ablation in mutant NPCs and suggested its potential involvement in the cilium.

To model WDR62 mutation-associated human microcephaly, we adopted a cerebral organoid culture system. Dual Smad-signaling inhibitors were added into neural induction medium to promote neuroepithelial expansion[35]. Embryoid bodies (EBs) were then transferred into droplets of Matrigel to promote complex tissue formation, followed by growth in a spinning bioreactor to enhance oxygen exchange and nutrient absorption (Supplementary Fig. 2A)[15]. To compare organoid formation of WDR62 mutant and isogenic controls, equal numbers (~9000 starting cells) of dissociated single PSCs were used to generate EBs, which exhibited indistinguishable morphology and surface areas at culture day 12 between controls and mutants. At week 4, control organoids developed large neuroepithelial loops that were persistent at week 5 and less visible at week 6; the overall organoid sizes consistently increased over time (Fig. 1d, Supplementary Fig. 3A). In contrast, the mutant cerebral organoids were drastically smaller in size and showed significantly reduced surface areas compared to controls (Fig. 1d, e). To confirm the phenotype specificity, we also generated cerebral organoids using two additional independent mutant hPSC clones (WDR62−/−-2 and WDR62−/−-3, Fig. 1a, b) with a long-term organoid culture method[17], starting with ~5000 PSCs. After 6 weeks of culture, different WDR62 mutations resulted in similar, smaller organoid sizes with reduced surface areas compared to controls (Supplementary Fig. 3B, C), suggesting the specificity of reduced cerebral organoid sizes from WDR62 mutations.

**Impaired NPC behaviors in mutant organoids.** MCPH is caused by the depletion of NPCs[1,3]. Previous studies revealed NPC reduction in Wdr62-deficient mouse models[12–14]. Therefore, we

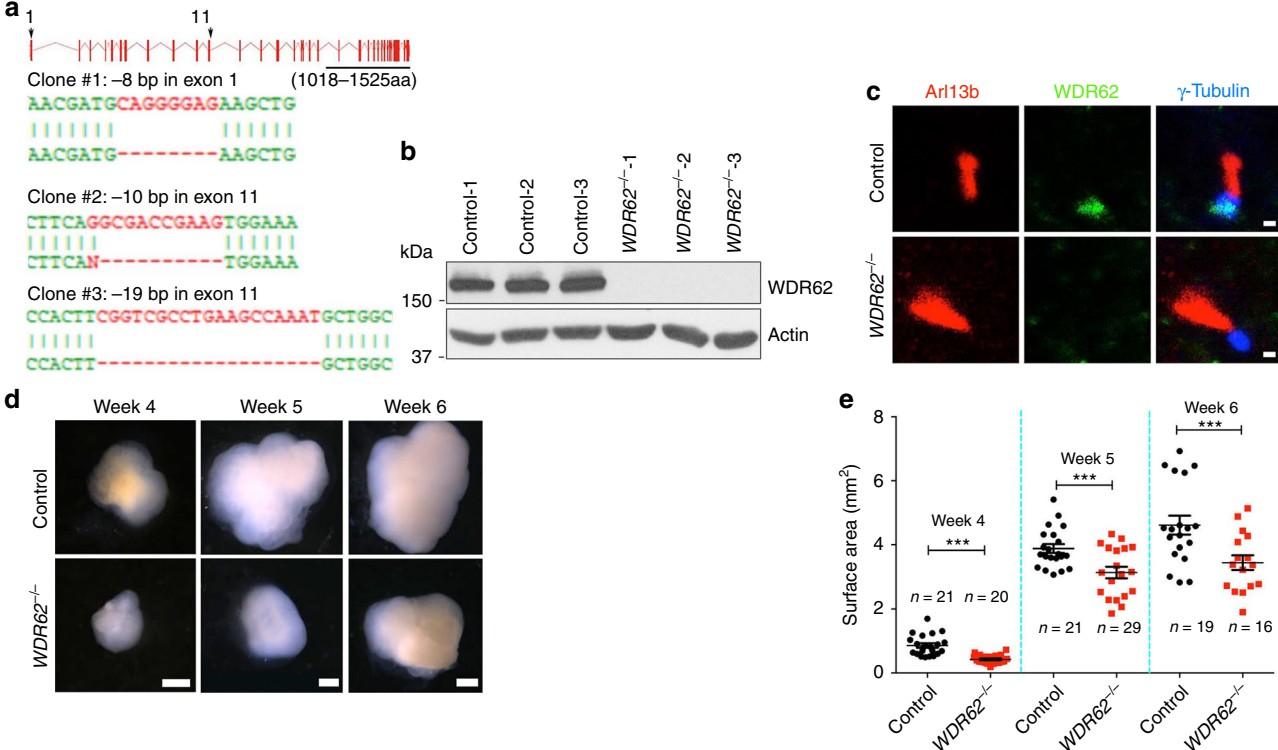

**Fig. 1** *WDR62* deletion results in smaller cerebral organoid sizes. **a**, **b** CRISPR/Cas9-mediated gene editing of human *WDR62* locus in pluripotent stem cells (PSCs), resulting in 8 bp depletion in exon 1, and 10 or 19 bp deletion in exon 11 (**a**). All of them resulted in ablation of WDR62 proteins due to premature mature stop codons (**b**). **c** Confocal imaging of control and mutant PSC-derived NPCs stained with antibodies against Arl13b (red), WDR62 (green), and γ-Tubulin (blue). Scale bars: 0.5 μm. **d** Representative images of control and *WDR62*$^{-/-}$ cerebral organoids at week 4, 5, and 6. Scale bars: 1 mm. **e** Quantification of surface areas of cerebral organoids. Error bars represent SEM of three independent experiments containing numbers of organoids as indicated; \*\*\**P* < 0.001 (Student's *t*-test)

examined NPCs and found a significant reduction of SOX2 and PAX6-labeled NPCs in mutant organoids (Fig. 2a, b, Supplementary Fig. 4A, B). In humans, oRGs are abundant in the cerebral cortex, but they are largely absent in rodent cortex, and are a primate-enriched progenitor population[6–8]. Therefore, we used SOX2 and PTPRZ1 co-labeling combined with their location in areas above VZ-like regions to identify oRGs in week 12 organoids (Fig. 2c)[16,18]. Ki67 labels cycling cells in all cell cycle phases except G$_0$. By quantifying Ki67-labeled cycling oRGs, we found a decrease in cycling oRGs in mutant organoids (Fig. 2d). These results suggest a NPC reduction in mutant organoids.

We previously reported a mitotic arrest defect in NPCs in a *Wdr62* KO mouse model[12]. Therefore, we examined mitosis and found an increase in p-H3-positive cells in VZ-like regions of mutant organoids (Fig. 2e, f). Using p-VIM to identify mitotic cells and Hoechst to mark nuclei (Fig. 2g), we analyzed the symmetric/asymmetric cell division pattern in mutant cerebral organoids. Quantification results showed that *WDR62* deletion resulted in an increase in vertical division, and a decrease in horizontal division in organoids (Fig. 2h), suggesting of impaired NPC division pattern. Sustained mitotic arrest leads to cell death[36,37]. Next, we examined cell death in cerebral organoids using TUNEL for late event of apoptosis in double strand DNA break and cleaved Caspase-3 staining for detecting early stage apoptosis, respectively. TUNEL staining results showed a significant increase in TUNEL-positive cells in mutant organoids (Fig. 2i, j). We also found an increase in cleaved Caspase-3-positive cells in mutant organoid sections (Fig. 2k, l). Together, these results suggest that *WDR62* deletion results in impaired mitosis, survival, and symmetric/asymmetric cell division in NPCs in the mutant cerebral organoids.

To study the consequence of NPC disruption, we combined apical neural progenitor markers PAX6/SOX2, IPC maker TBR2, and the deep-layer cortical marker CTIP2. There we identified VZ-, SVZ-, and CP-like regions in organoids that resembled human cortical development. VZ is revealed by dense PAX6-positive cells towards ventricle sides; SVZ is less dense and marked with TBR2 expression; and CP is immediately superior to the TBR2 boundary coupled with CTIP2 expression. We analyzed IPCs marked by TBR2 expression and found that there was a significant decrease in TBR2-positive cells in mutant organoids (Supplementary Fig. 4C, D). Consistent with reduced aRGs and IPCs, there was a corresponding reduction of CTIP2-positive cortical neurons in mutant organoids (Supplementary Fig. 4C, E). Therefore, *WDR62* deletion disrupts human NPC behaviors and leads to NPC and neuron depletion, which results in smaller organoid sizes.

**Wdr62 knockout mice exhibited smaller brains with reduced NPCs**. Our previously reported *Wdr62*-deficient mice were gene trap lines with residual proteins and exhibited subtle microcephaly phenotypes[12]. These data raised the question whether subtle microcephaly could be due to the residual protein. To completely remove Wdr62 protein, we generated a conditional allele through homologous recombination followed by germline transmission (*Wdr62*$^{f/f}$, Fig. 3a). Sequence analysis validated the correct locations of Frt and LoxP sites in the *Wdr62* gene (Supplementary Fig. 5A,B). After crossing a CMV-Cre line with *Wdr62*$^{f/f}$ mice, a null allele was generated due to a premature stop codon in the *Wdr62* gene after removing exon 2 (Fig. 3a). Western blot analyses confirmed the absence of Wdr62 protein in

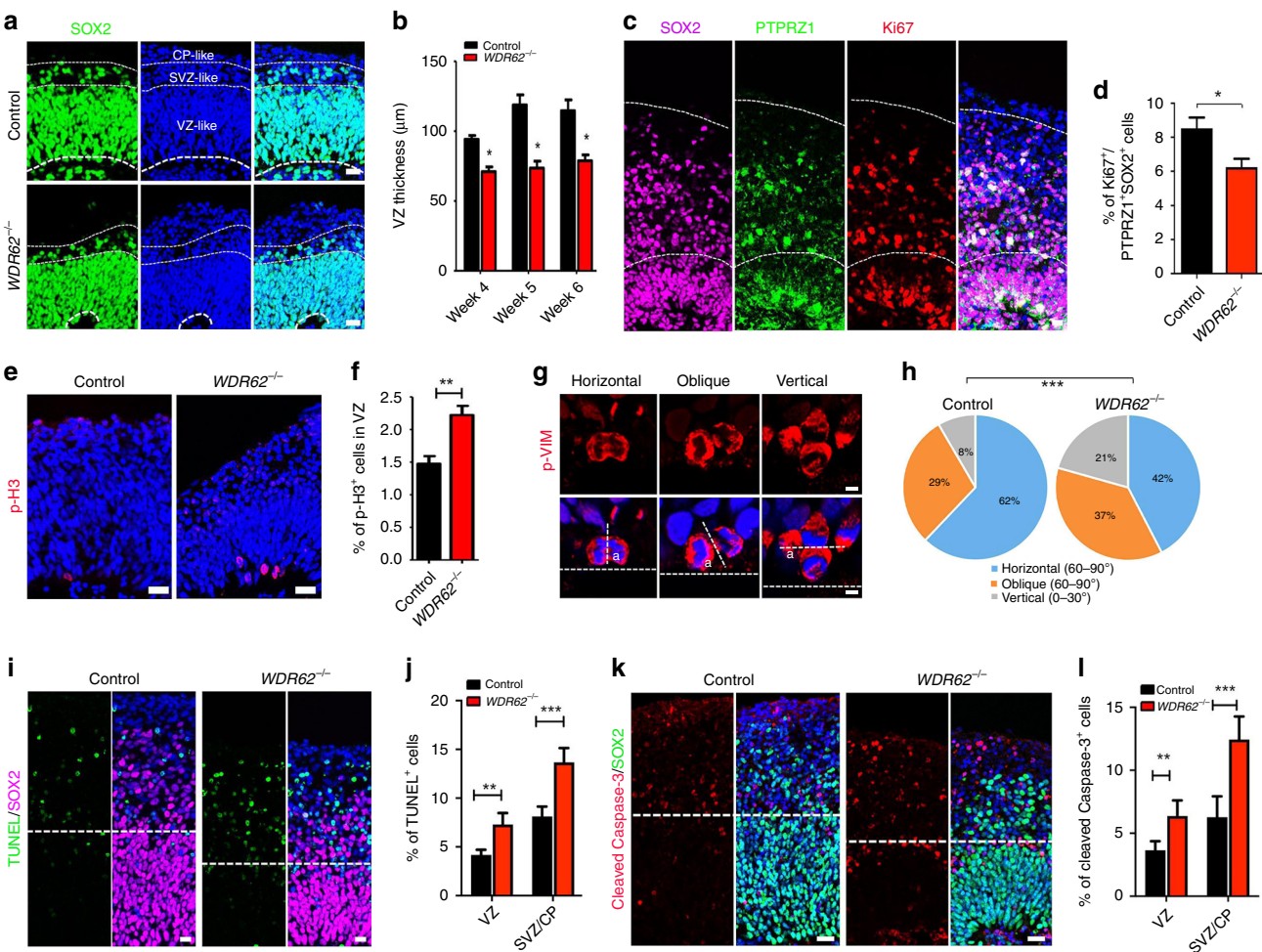

**Fig. 2** Impaired mitosis, symmetric division, survival of NPCs in mutant organoids. **a** Confocal imaging of immunohistochemical (IHC) staining sections from week 6 cerebral organoids. Scale bars: 20 μm. White dotted lines mark ventricular zone (VZ), subventricular zone (SVZ), and cortical plate (CP)-like areas. **b** Quantification of SOX2-positive VZ thickness in organoids. **c** Representative imaging of IHC staining from week 12 control cerebral organoids. Regions below bottom white dotted lines represent VZ/SVZ-like regions. Hoechst stains nuclei (blue). Scale bars: 20 μm. **d** Quantification of Ki67-positive out of total PTPRZ1- and SOX2-double positive cells (*n* > 50 cells counted in each experiment). **e**, **g**, **k** Representative imaging of IHC staining of week 6 organoid sections with antibodies listed. Above and below white dotted lines in K represent SVZ/CP- and VZ-like regions, respectively. Hoechst stains nuclei (blue). Scale bars: 20 μm in (**e**), 5 μm in (**g**), and 10 μm in (**k**). **f**, **h**, **l** Quantification of the percentage of p-H3 (**f**), division angle (**h**), and percentage of Caspase-3-positive cells in the VZ-like or SVZ/CP-like regions in week 6 organoid sections. **i** Representative imaging of TUNEL staining in week 6 organoid sections. **j** Quantification of the percentage of TUNEL-positive cells. Error bars represent SEM of results from three independent cerebral organoids (three sections from each organoid); *P < 0.05, ***P < 0.001 (Student's t-test). In **h**, ***P < 0.001 (Mann–Whitney test)

mutant cells (Fig. 3b). *Wdr62* heterozygous mice were viable and fertile with indistinguishable size from littermate controls.

Smaller brain sizes and brain weights were initially detected in *Wdr62* homozygous knockout (KO) at embryonic day 18.5 (E18.5) (Fig. 3c, d). We also examined postnatal day 30 (P30) brains and confirmed that mutant brains were smaller than controls (Fig. 3e). Hematoxylin and eosin (H&E) staining revealed a significant decrease in cortical radial thickness in mutants (Fig. 3f, g). Interestingly, complete ablation of Wdr62 also led to mild microcephaly, in contrast to the drastic reduction of mutant organoid sizes (Fig. 1d, e). To determine whether the reduction in brain size was due to reduced NPC numbers, we examined apical neural progenitor cells (APCs) and IPCs labeled by Pax6 and Tbr2, respectively. We used coronal cerebral sections obtained from similar rostral–caudal axes from E16.5 and E18.5 brains. At both stages, the numbers of Pax6-positive APCs and Tbr2-positive IPCs were significantly decreased in the mutants (Fig. 3h–k). Further analyses showed that there was a significant increase in phospho-histone 3 (p-H3)-positive NPCs in the

mutant VZ/SVZ (Supplementary Fig. 5C, D). Increased apoptotic cells in mutant cortex were validated by TUNEL assay (Supplementary Fig. 5E, F). Overall, these results suggest that WDR62 depletion causes reduced cortical sizes, which appears more severe in human cerebral organoids than that in mouse brains.

**Increased cilium length in *WDR62* mutant NPCs.** The primary cilium consists of axoneme microtubules that extend from the basal body, which is derived from the mother centrioles of centrosomes[2,38]. WDR62 is a centrosome protein with functions in centriole biogenesis[12,13,30]. We found that WDR62 localizes to the basal body of the primary cilium in human NPCs, revealed by its co-labeling with Arl13b and γ-tubulin (Fig. 1c).

We hypothesized that WDR62 may play a role in ciliogenesis in NPCs. To test this hypothesis, we examined cilium numbers and morphology in asynchronously proliferating human NPCs. Around 20% of wild type human NPCs displayed cilia labeled

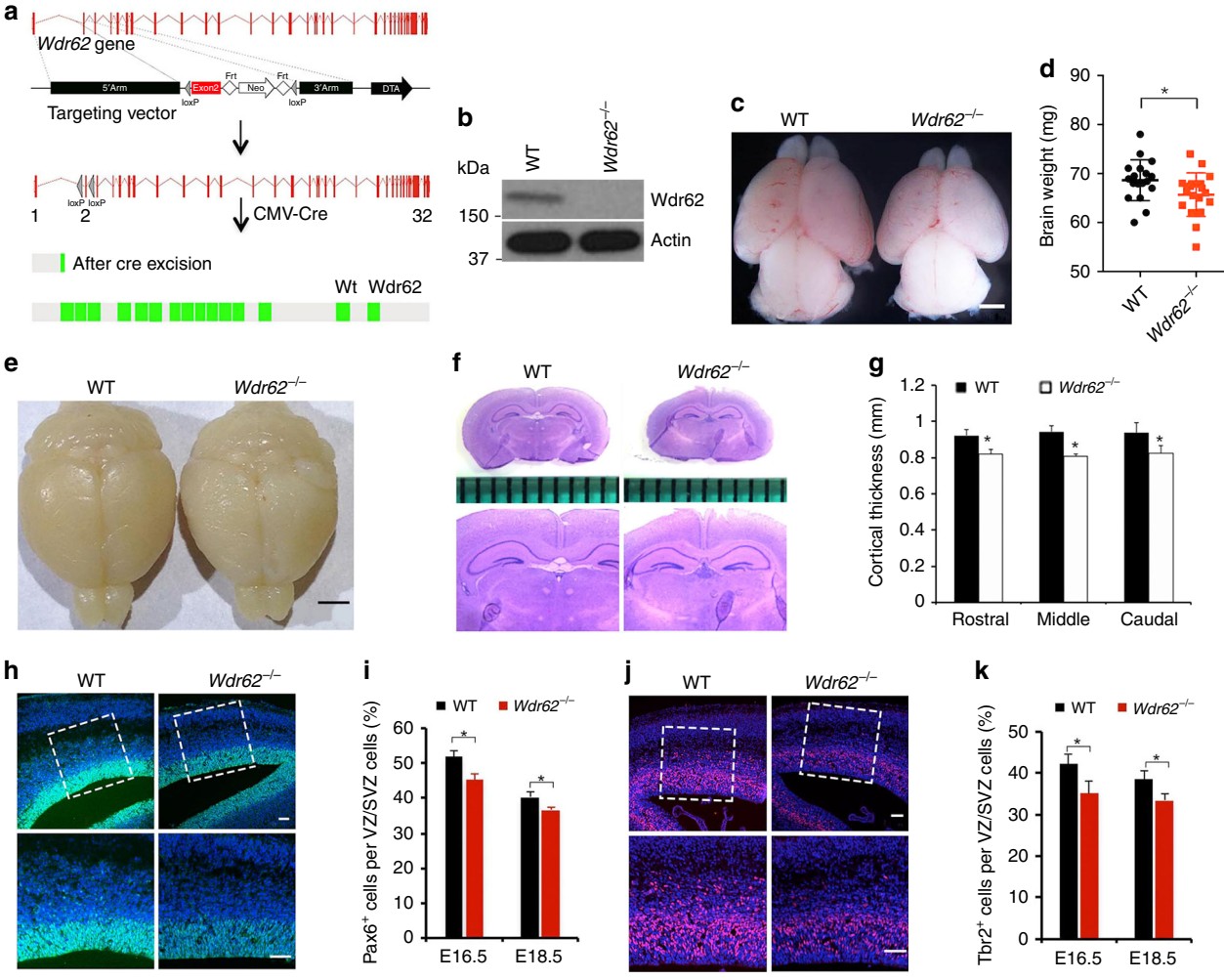

**Fig. 3** *Wdr62* deletion in mice results in smaller brain size and reduced NPCs. **a** Schematic diagram of strategy for *Wdr62* null mouse generation. **b** Western blot analyses of Wdr62 protein expression in mouse embryonic fibroblasts (MEFs) isolated from E14.5 embryos. β-actin serves as the control. **c**, **e** Dorsal views of E18.5 and P30 mouse brains. Scale bars: 1 and 2 mm. **d** Statistical analysis of E18.5 WT (*n* = 17) or mutant (*n* = 17) brain weights. *P < 0.05 (Student's *t*-test). **f** Coronal sections of P30 cerebral cortex stained with haematoxylin and eosin. The lower panels are the enlargements of upper panels . **g** Quantification of radial thickness in neocortex. Error bars indicate SEM of 9 sections at comparable positions along the rostral/caudal axis of cerebral cortex from three independent experiments. *P < 0.05 (Student's *t*-test). **h**, **j** Confocal microscope images of coronal sections of E16.5 cerebral cortex stained with antibodies against Pax6 (green, **h**) or Tbr2 (red, **j**). Hoechst stains nuclei (blue). Lower panels are the enlargements from the white boxed areas in the upper panels. Scale bars: 50 μm. **i**, **k** Quantification of the percentage of Pax6-positive and Tbr2-positive cells per total number of NPCs within VZ/SVZ. Error bars represent SEM of 9 sections from three independent experiments. *P < 0.05 (Student's *t*-test)

with Arl13b. In contrast, there was a significant increase in the percentage of ciliated NPCs after *WDR62* deletion (Fig. 4a, b). We measured cilium length in individual NPCs and found that mutant NPCs contained unusually long cilia with a median length of 2.8 ± 0.6593 μm compared to 2.09 ± 0.6584 μm in control cells (Fig. 4c, d). Increased cilium length occurred in NPCs from multiple independent mutant clones (Supplementary Fig. 3D, E). To determine whether aberrant ciliogenesis of mutant NPCs occurred ex vivo, we analyzed cerebral organoids. We used Arl13b and γ-tubulin to label cilia and centrosomes, respectively. Focusing on NPCs located in VZ-like regions, we found that the median length of primary cilia was significantly increased in mutants compared to controls (Fig. 4e, f). Consistent with 2D-cultured NPCs, there was also a significant increase in the percentage of ciliated NPC numbers in mutant organoids (Fig. 4e, g). Together, these results suggest that *WDR62* deletion led to substantially increased cilium length and numbers in human NPCs.

To determine whether Wdr62 regulates ciliogenesis in mice in vivo, we examined NPCs in the developing brain. We analyzed

NPCs in the VZ of the cerebral cortex at E10.5 stage. Primary cilia were labeled by co-staining using antibodies against acetylated α-tubulin and γ-tubulin (Fig. 4h). There was a slight but significant increase in the median length of primary cilia in mutant NPCs (Fig. 4h, i). The cilium number was also slightly increased in mutant cerebral cortex (Fig. 4j). To further confirm the increased cilium length in mutant brains, we performed scanning electron microscopy (SEM) analysis. There was an increase in cilium length in mutant NPCs in the developing cortex (Fig. 4k, l), which was statistically significant (Fig. 4m). Together, these results suggest a mild increase in cilium length and numbers in mutant NPCs in the developing mouse brain. In contrast, these cilium deficits are robust in mutant human NPCs and organoids.

**Impaired cilium disassembly and cell cycle progression in mutant cells.** The high percentage of ciliated cells observed in mutant NPCs could be due to increased cilium formation and/or decreased cilium disassembly, which cannot be distinguished in cultured human or mouse NPCs. Therefore, we turned our

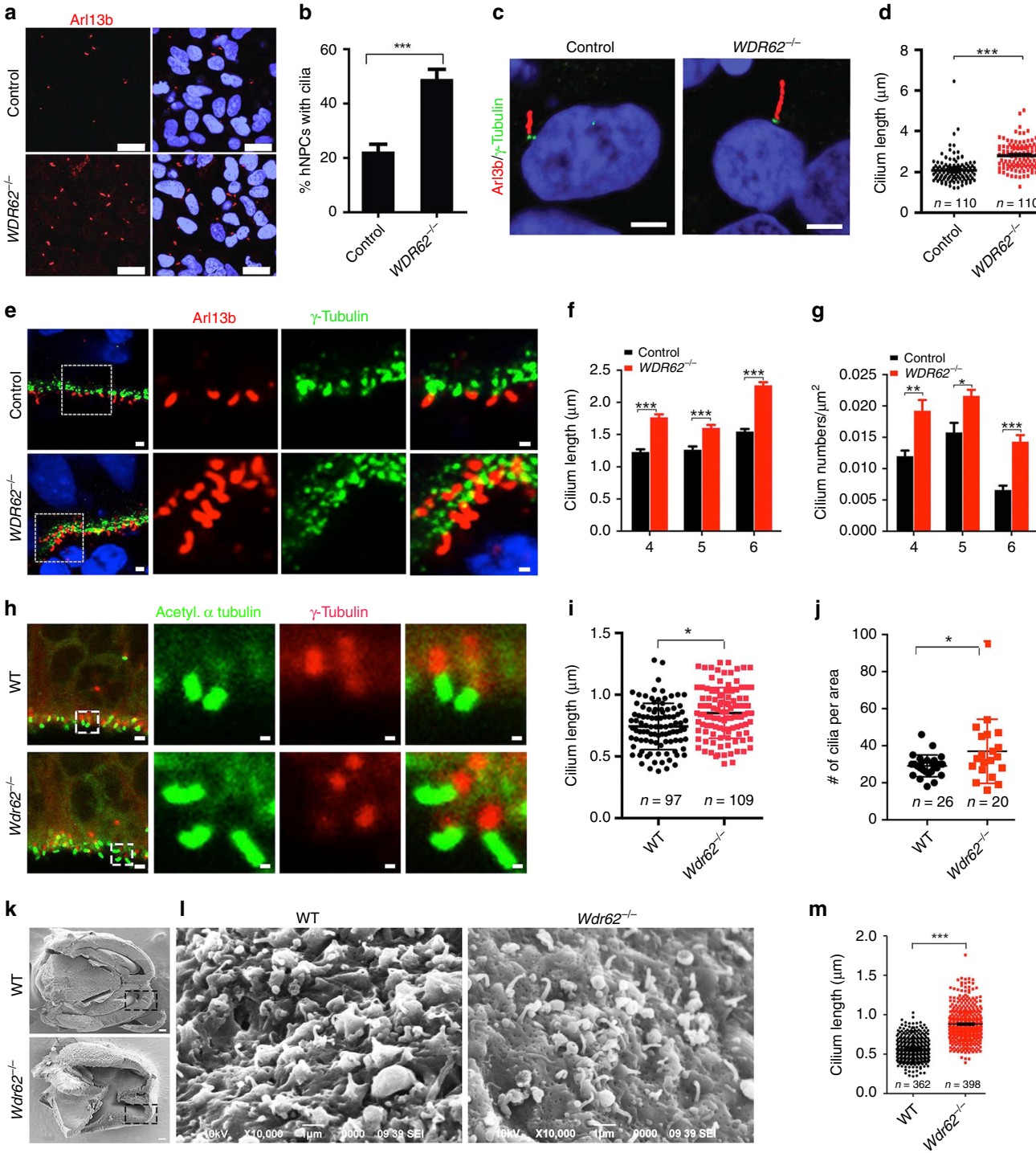

**Fig. 4** Increased cilium length in mutant NPCs in cerebral organoids and mice. **a** Confocal imaging of 2D human NPCs stained with antibodies against Arl13b (red). Hoechst stains nuclei (blue). Scale bars: 50 μm. **b** Quantification of the percentage of human NPCs with cilium. Error bars represent SEM of three independent experiments from control (*n* = 168) and mutant (*n* = 186) NPCs; ***P < 0.001 (Student's *t*-test). **c** Confocal imaging of 2D human NPCs stained with antibodies against Arl13b (red) and γ-tubulin (green). Hoechst stains nuclei (blue). Scale bars: 2.5 μm. **d** Quantification of cilium length in NPCs. Error bars represent SEM of three independent experiments from control (*n* = 110) and mutant (*n* = 110) NPCs with cilia; ***P < 0.001 (Student's *t*-test). **e** Confocal imaging of sections from week 6 organoids stained with antibodies against Arl13b (red) and γ-tubulin (green). Hoechst stains nuclei. Scale bars: 1 μm. **f, g** Quantification of cilium length or numbers per μm² in organoid sections. Error bars represent SEM of three independent experiments; *P < 0.05, **P < 0.01, ***P < 0.001 (Student's *t*-test). **h** Confocal imaging of coronal sections of E12.5 cerebral cortex stained with antibodies against acetylated α-tubulin (green) and γ-tubulin (red). Scale bars: leftmost, 2 μm; right, 0.35 μm. **i, j** Quantification of cilium length and numbers per area (598.148 μm²) in E10.5 developing cortex. Error bars represent SEM of three independent experiments; *P < 0.05 (Student's *t*-test). **k, l** Scanning electron microscopy (SEM) analysis of cilium in E12.5 cortex. Panels in **l** are enlargement of boxed areas in **k**. Scale bars: 100 μm in **k**, and 1 μm in **l**. **m** Quantification of cilium length from WT (*n* = 362) and mutant (*n* = 398) SEM results in **l**

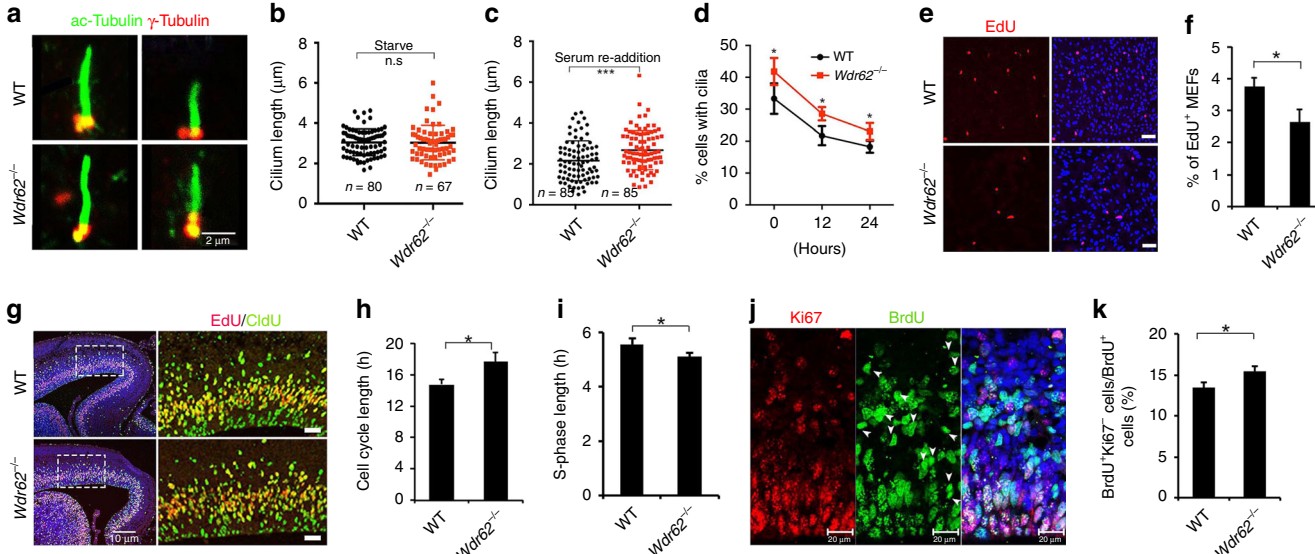

**Fig. 5** Delayed cilium disassembly, cell cycle re-entry, and extended cell cycle length in mutant cells. **a** MEFs stained with antibodies against acetylated α-tubulin (green) and γ-tubulin (red). Scale bars: 2 μm. **b** Quantification of cilium length of WT ($n = 80$) and mutant ($n = 67$) MEFs serum starved for 24 h. **c** Quantification of cilium length of WT ($n = 85$) and mutant ($n = 85$) MEFs after serum re-addition for 24 h. **d** Quantification of the percentage of cells with cilia. MEFs were serum-starved for 24 h followed by re-stimulation with serum for 0, 12, or 24 h. **e** Representative imaging of MEFs with EdU staining (red). Hoechst stains nuclei. Scale bar: 100 μm. **f** Quantification of the percentage of EdU-positive cells out of total cells counted (~200 cells in each experiment). **g** Coronal sections of E16.5 cerebral cortex stained with EdU (red) and antibodies against CldU (green). Hoechst stains nuclei (blue). Right panels are enlargements of regions outlined by white dotted boxes in leftmost panels. Scale bar: 10 μm (leftmost panels); 20 μm (right panels). **h, i** Quantification of cell cycle length and S-phase length. **j** Representative imaging of E16.5 cortex sections stained with antibodies against Ki67 (red) and BrdU (green) after 24-h BrdU labeling. White arrowheads represent BrdU-positive and Ki67-negative cells. **k** Quantification of BrdU-postive and Ki67-negative cells out of total BrdU-positive cells. Error bars represent SEM of three independent experiments. *$P < 0.05$, ***$P < 0.001$, n.s represents not significant (Student's $t$-test)

attention to MEFs, which have been routinely used to examine cilium assembly and disassembly[23–25]. Serum starvation in MEFs induces cilium assembly, whereas growth stimulation by serum re-addition drives cilium disassembly (Fig. 5a)[38]. We synchronized MEFs at G0 and induced cilium formation by serum starvation. We found that cilia in both WT and mutant MEFs acquired similar length (Fig. 5b), which suggests normal cilium assembly in mutant MEFs. In contrast, mutant MEFs displayed an increase in cilium length at 24 h after serum re-addition (Fig. 5a, c). By examining large numbers of MEFs, we found that the percentage of ciliated MEFs was significantly higher in mutants at multiple time points after serum re-addition (Fig. 5d). Together, these results suggest that *Wdr62* deletion caused a delay of cilium disassembly resulting in more ciliated MEFs.

An elongated cilium is associated with suppression of cell cycle progression via delaying G1-S phase transition[22–24]. We therefore examined the effect of *Wdr62* deletion on cell cycle re-entry. MEFs were synchronized by serum starvation for 24 h followed by growth stimulation to drive cell cycle re-entry, during which DNA-replicating cells were identified by pulse-labeling experiments using ethynyl-deoxyuridine (EdU). There was a significant decrease in the percentage of EdU-positive cells in mutant MEFs (Fig. 5e, f). A disruption of cell cycle progression would extend cell cycle length. We performed cell cycle length analyses of NPCs localized in the VZ of mutant cerebral cortex (Fig. 5g). Mutant NPCs exhibited an increase in cell cycle length (Fig. 5h), and a decrease in S-phase length (Fig. 5i). To determine whether impaired cell cycle progression is coupled with precocious differentiation, we performed 24 h BrdU labeling and found there is a significant increase in BrdU-positive and Ki67-negative cells (Fig. 5j, k), suggesting a premature cell cycle exit of mutant NPCs. Together, these data suggest that *Wdr62* depletion delayed

cilium disassembly and cell cycle re-entry and leads to extended cell cycle length and precocious cell cycle exit of NPCs.

**Impaired proliferation and differentiation of mutant human NPCs.** To determine cell cycle progression in human NPCs, we used hPSC-derived NPCs within three passages, which maintain a dorsal cortical identity as revealed by PAX6 staining (Fig. 6a, b). The 2-h BrdU-labeling studies showed that there was a significant decrease in the percentage of BrdU-positive mutant NPCs compared to controls (Fig. 6c, d). The percentage of Ki67-positive human NPCs was also reduced (Fig. 6e, f). Therefore, *WDR62* deletion resulted in a decrease in proliferation of mutant human NPCs. These in vitro results, coupled with the reduced NPC pool in mutant organoids, prompted us to examine the effect of *WDR62* deletion on NPC proliferation in cerebral organoids. Both control and mutant brain organoids exhibited stereotypical organization patterns of neuroepithelial cells reminiscent of the developing cerebral cortex, where BrdU-labeled DNA replicating NPCs and Ki67-marked cycling NPCs are located in the VZ of cerebral organoids (Fig. 6g). There was a significant decrease in the percentage of BrdU-positive NPCs in mutant organoids at weeks 4 and 5, but not 6 (Fig. 6g, h). Compared to WT cells, mutant organoids contained significantly fewer Ki67-positive NPCs (Fig. 6g, i). Therefore, *WDR62* deletion caused a significant decrease in human NPC proliferation.

Altered G1-S phase transition impacts the balance between proliferation and differentiation of NPCs[39]. These studies above showed a delayed cell cycle re-entry and G1-S phase transition in *Wdr62* mutant MEFs and disrupted cell cycle progression of mutant NPCs. Therefore, we reasoned that the reduced NPC proliferation in mutant organoids is coupled with changes in their

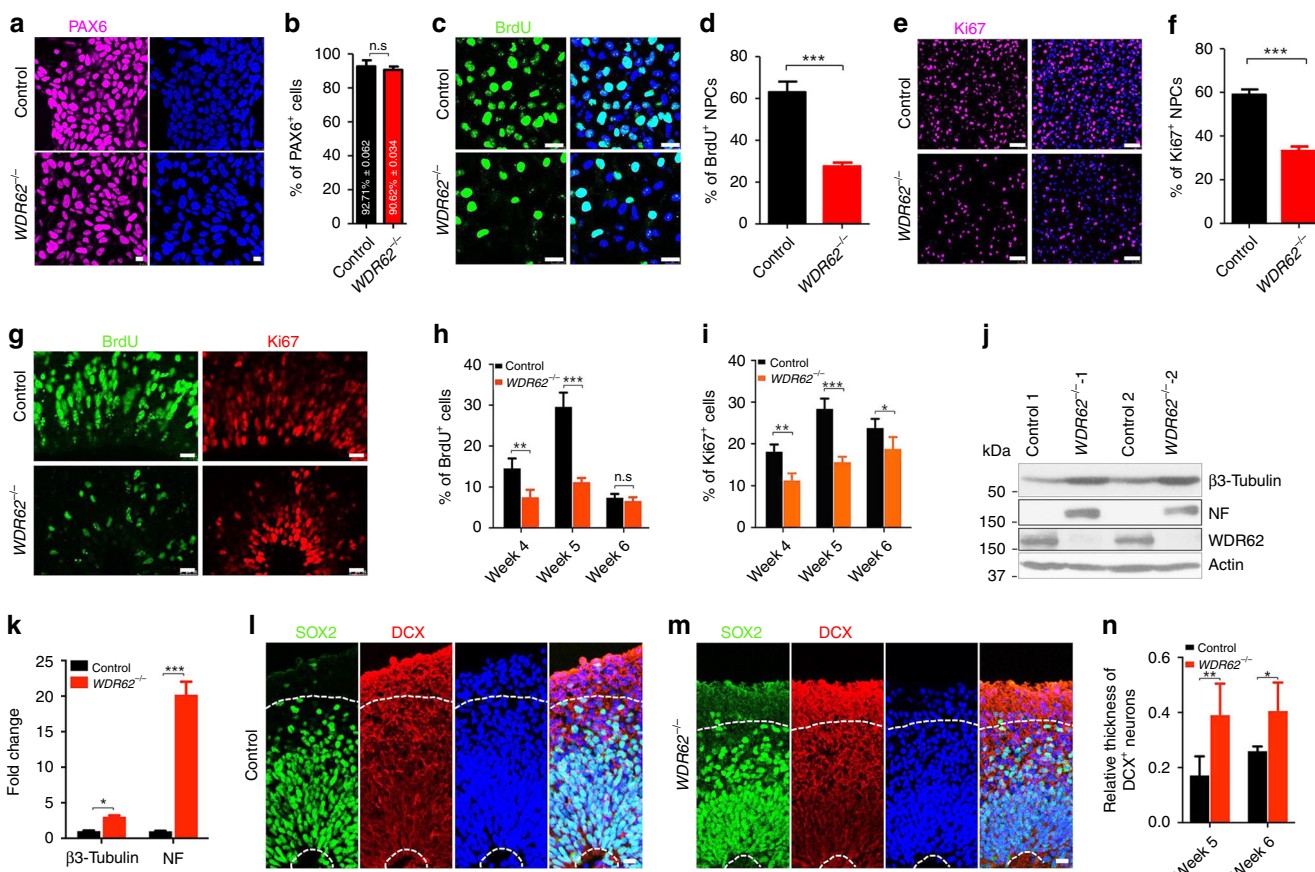

**Fig. 6** Decreased proliferation and premature differentiation of human NPCs. **a, c, e** Representative imaging of 2D human NPCs stained with antibodies against PAX6, BrdU (after 2-h BrdU pulse), or Ki67. Hoechst stains nuclei. Scale bars: 10 μm in **a**, 25 μm in **c**, and 50 μm in **e**. **b, d, f** Quantification of the percentage of PAX6-positive, BrdU-positive, or Ki67-positive cells out of total NPCs (~200 cells counted in each experiment). **g** Confocal imaging of week 5 organoid sections stained with antibodies against BrdU (green) and Ki67 (red). Hoechst stains nuclei (blue). Scale bars: 20 μm. **h, i** Quantification of the percentage of BrdU-positive or Ki67-positive cells out of total cells in VZ-like regions in cerebral organoids. **j** Western blot analysis of protein levels of β3-Tubulin and Neurofilament (NF) in 2D human NPCs. **k** Quantification of western blot data from three independent experiments. **l, m** Representative imaging of week 6 organoid sections stained with antibodies against SOX2 (green) and DCX (red). Hoechst stains nuclei. Areas above upper white dotted lines represent DCX-positive neuron regions for quantification. Scale bars: 20 μm. **n** Quantification of relative thickness of DCX-positive neurons. Error bars represent SEM of three independent experiments. *P < 0.05; **P < 0.01; ***P < 0.001; n.s. represents no significant difference detected (Student's t-test)

differentiation propensity. Western blot analyses showed that mutant cells exhibited an increase in the expression of β3-Tubulin and neurofilament (NF), two neuronal differentiation markers (Fig. 6j, k). Next, we examined cerebral organoids. At week 4, neuronal differentiation was initiated in cortical plate (CP)-like regions of cerebral organoids, revealed by DCX staining (Supplementary Fig. 2C). There was a robust increase in the percentage of DCX-positive neurons in CP-like regions in mutant organoids (Fig. 6l–n). Together, WDR62 deletion caused a reduced proliferation and a precocious differentiation of human NPCs.

**WDR62 recruits CEP170 to the basal body.** To investigate the molecular mechanisms by which WDR62 regulates cilium disassembly, we developed a human embryonic kidney (HEK) 293 cell-derived stable cell line expressing Flag-Wdr62 protein. Immunostaining results revealed that Flag-Wdr62 localized to the spindle poles (Supplementary Fig. 6A), resembling its endogenous expression in human NPCs (Supplementary Fig. 1). The cytoplasmic extract enriched for the Flag-Wdr62 protein was affinity-purified using Flag resin followed by size-exclusion chromatography. Analyses of column fractions after silver staining suggested that Wdr62 protein co-eluted with proteins

in fractions 16–24 and fractions 26–30 (Supplementary Fig. 6B). The individual bands were extracted from the gel and subjected to mass spectrometry. This analysis identified CEP170 as having the closest peptide coverage to that of Wdr62. Co-immunoprecipitation (Co-IP) analyses confirmed that Cep170 interacts with Wdr62 (Fig. 7a).

Cep170 is a centrosome protein serving as a marker for mature centrioles[40], but its potential involvement in cilia has not been reported. We found that Cep170 co-localizes with Wdr62 at the basal body of the primary cilium (Fig. 7b), whereas its total protein levels were not changed in mutant MEFs (Supplementary Fig. 6C, D). Wdr62 is a WD-repeat protein with a scaffold function[41]. We found that there was a significant decrease of Cep170 localization at the basal body of the primary cilium in mutant human NPCs (Fig. 7c, d). A similar reduction of basal body localization of Cep170 was also observed in mutant MEFs (Supplementary Fig. 7A, B). Next we tested whether the converse is also true by determining whether Cep170 is required for Wdr62's basal body localization. CEP170 depletion did not affect WDR62's localization at the basal body of the primary cilium (Supplementary Fig. 7C, D). Therefore, these results suggest that WDR62 recruits CEP170 to the basal body of the primary cilium, rather than vice versa.

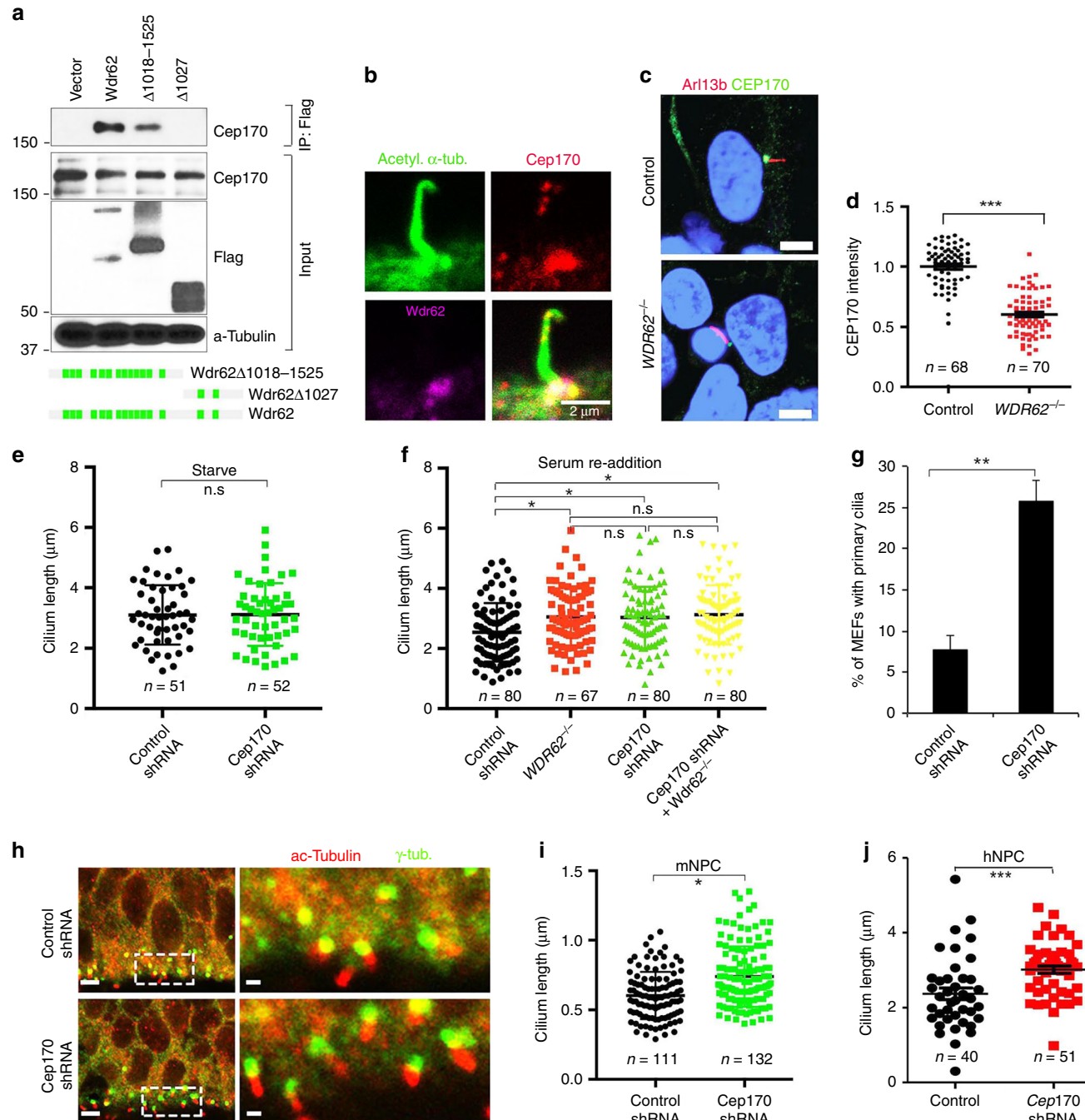

**Fig. 7** WDR62 recruits CEP170 to the basal body for promoting cilium disassembly. **a** Cep170 interacts with Wdr62 by co-IP assay. Bottom panels: schematic diagrams of full length, N-terminal, and C-terminal *Wdr62* constructs. **b** MEFs stained with antibodies against acetylated α-tubulin (green), Cep170 (red), and Wdr62 (purple). Scale bars: 2 μm. **c** 2D human NPCs stained with antibodies against Arl13b (red) and CEP170 (green). Scale bars: 2.5 μm. **d** Quantification of CEP170 signal intensity in control ($n = 68$) and mutant ($n = 70$) human NPCs with cilia. **e** Quantification of cilium length of control ($n = 51$) and Cep170 knockdown ($n = 52$) MEFs under serum starvation condition. **f** Quantification of cilium length of MEFs, including WT ($n = 80$), *Wdr62*$^{-/-}$ ($n = 67$), *Cep170* knockdown ($n = 80$), and *Cep170* knockdown in *Wdr62*$^{-/-}$ ($n = 80$) MEFs with cilia, after serum re-addition for 24 h. **g** Quantification of the percentage of ciliated MEFs out of total cells (~500 cells) at 24 h after serum re-addition. **h** Coronal sections of E16.5 cerebral cortex stained with antibodies against acetylated α-tubulin (red) and γ-tubulin (green). Control or *Cep170* shRNA together with H2B-GFP constructs were injected into E14.5 ventricles. Right panels are enlargements of regions outlined by white dotted boxes in left panels. Scale bar: 5 μm (left panels); 1 μm (right panels). **i** Quantification of cilium length within a $1.153 \times 10^5$ μm$^2$ field in **h**. **j** Quantification of cilium length in WT ($n = 40$) and *CEP170* knockdown ($n = 51$) human NPCs. Error bars represent SEM of three independent experiments. *$P < 0.05$; **$P < 0.01$; ***$P < 0.001$; n.s. represents no significant difference detected (Student's *t*-test)

Next, we examined whether CEP170 and WDR62 have similar functions in cilium disassembly. Cilium formation appeared normal in Cep170-depleted MEFs (Fig. 7e). However, there was a significant increase in cilium length in Cep170-depleted MEFs upon serum re-addition (Fig. 7f), which resembles the cilium disassembly defect in *Wdr62*-mutant cells. Furthermore, Cep170 depletion in the background of mutant cells did not exacerbate the cilium length defect observed in *Wdr62* null MEFs (Fig. 7f), suggesting that they may not function in parallel. Similar to *Wdr62*-mutant MEFs, the percentage of ciliated MEFs was significantly increased upon Cep170 depletion (Fig. 7g). Overall, these results suggest that these two proteins likely function in the same pathway to promote cilium disassembly, which is consistent with Cep170's basal body recruitment by Wdr62. To determine whether Cep170 regulates ciliogenesis in NPCs in vivo, we performed in utero electroporation to knock down its expression in the developing mouse cortex. Knockdown cells were co-labeled with H2B-GFP and exhibited an increase in the proportion of the p-H3-positive NPCs in the VZ of the developing cortex (Supplementary Fig. 7E, F), which is similarly found in the *Wdr62*

mutant brains (Supplementary Fig. 5C, D). Average cilium length was significantly increased in NPCs at the VZ of the developing cerebral cortex upon Cep170 depletion (Fig. 7h, i). Cilium length was also increased in *CEP170* knockdown human NPCs (Fig. 7j). Together, these results suggest that CEP170 is a potent downstream mediator of WDR62 function in cilium disassembly.

**The rescue effects of KIF2A overexpression in organoids.** KIF2A, a kinesin-13 family member, promotes cilium disassembly and interacts with CEP170[26,42]. Therefore, we hypothesized that CEP170 recruits KIF2A to the basal body of the primary cilium. To test this hypothesis, we examined KIF2A localization and found that there was a significant decrease of KIF2A basal body localization due to CEP170 depletion (Fig. 8a, b). These data suggest that CEP170 is required for KIF2A's appropriate localization in human NPCs. Furthermore, we found a reduction of KIF2A basal body localization in primary cilia of *WDR62* mutant NPCs (Fig. 8c, d). To determine the specificity of KIF2A downregulation, we examined components critical for cilium assembly, including the initiation factor Cep164

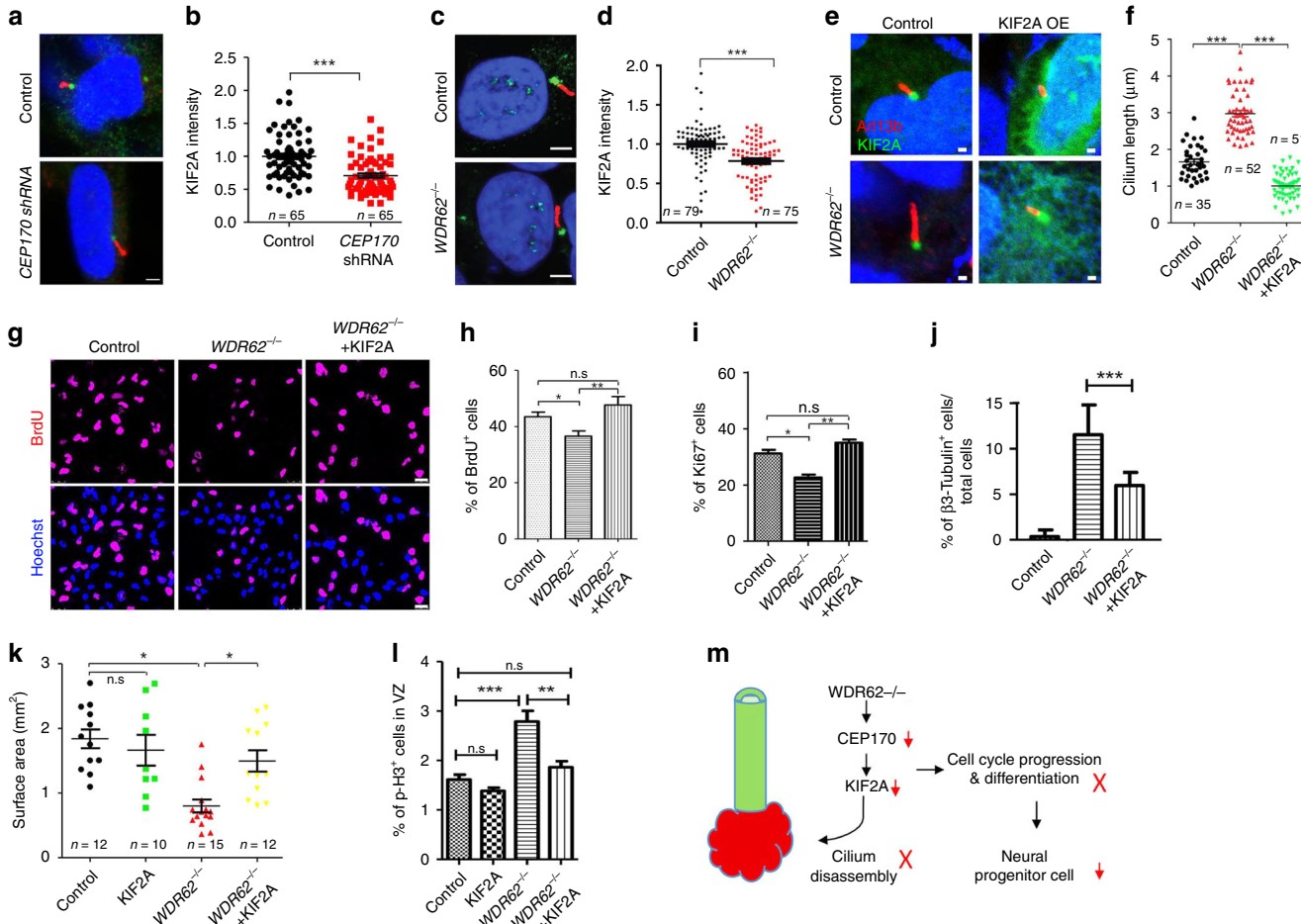

**Fig. 8** Enhanced KIF2A expression partially rescued deficits in cilium length, NPC proliferation, and organoid sizes. **a, c** Confocal imaging of *CEP170* knockdown (**a**) or *WDR62* mutant (**c**) human NPCs stained with antibodies against Arl13b (red) and KIF2A (green). Hoechst stains nuclei (blue). Scale bars: 2.5 μm. **b, d** Quantification of KIF2A signal intensity. **e** Confocal imaging of control or KIF2A-GFP lentivirus-infected human NPCs stained with antibodies against Arl13b (red) and KIF2A (green). Hoechst stains nuclei (blue). Scale bars: 1 μm. **f** Quantification of cilium length in **e**. **g** Confocal imaging of BrdU-positive human NPCs (red). Hoechst stains nuclei (blue). Scale bars: 25 μm. **h–j** Quantification of the percentage of BrdU, Ki67, and B3-Tubulin-positive cells out of total human NPCs. **k** Quantification of surface areas of cerebral organoids. **l** Quantification of p-H3-positive cells in VZ-like regions of mutant organoids with or without KIF2A overexpression. For all the experiments, error bars represent SEM of three independent experiments. *P < 0.05, **P < 0.01, ***P < 0.001, n.s. represents not significant (Student's t-test). **m** Diagram showing that a WDR62-CEP170-KIF2A microcephaly protein pathway promotes cilium disassembly, disruption of which reduces NPCs and contributes to microcephaly

and two components of the intraflagellar transport (Ift) complex, Ift81 and Ift88[43–45]. Immunostaining results showed that their localization and expression levels were not significantly changed in mutant MEFs (Supplementary Fig. 8). Together, these results suggest that WDR62 recruits CEP170-KIF2A to the basal body of primary cilia in NPCs.

Mutations in *KIF2A* cause MCPH in humans[46]. KIF2A acts in a microtubule-depolymerizing activity-dependent manner to drive cilium disassembly, which is linked to cell proliferation and cortical malformation[26,47]. KIF2A's basal body localization was reduced in *WDR62*-deficient or *CEP170*-deficient NPCs (Fig. 8a–d). Therefore, we reasoned that WDR62 drives cilium disassembly by promoting CEP170's localization to the basal body of the primary cilium, where CEP170 recruits KIF2A to disassemble the cilium. To test this hypothesis, we overexpressed KIF2A and examined whether the unusually long cilia in *WDR62* mutant NPCs can be rescued. Lentivirus-mediated KIF2A overexpression was detected at the basal body of the primary cilium (Fig. 8e). Statistical analysis revealed that KIF2A over-expression rescued the cilium length defect in *WDR62*-mutant human NPCs (Fig. 8e, f). Next, we examined whether the proliferation defect in mutant NPCs could be rescued by KIF2A overexpression. BrdU/Ki67-labeling studies revealed that KIF2A rescued the reduced proportion of S-phase cells and total cycling NPCs (Fig. 8g–i). The precocious differentiation of *WDR62*-mutant NPCs was partially rescued by KIF2A over-expression (Fig. 8j). Importantly, KIF2A overexpression partially rescued the organoid size defects of *WDR62* mutants (Fig. 8k, Supplementary Fig. 5A–C). The increased p-H3-positive cells was also reduced by KIF2A overexpression in cerebral organoids (Fig. 8l, Supplementary Fig. 9D). These results indicate that WDR62 has a KIF2A-mediated function in cilium regulation that is essential for NPC maintenance and cortical development. Therefore, we identified a WDR62–CEP170–KIF2A pathway promoting cilium disassembly, disruption of which impairs NPC maintenance and contributes to MCPH (Fig. 8m).

## Discussion

Although NPC disruption is known as the cause of MCPH[1,9], genetic mouse models cannot model the severity of human MCPH, suggesting that certain aspects of MCPH gene functions cannot be fully revealed in mice. Therefore, the cellular and molecular mechanisms underlying MCPH remain elusive, in particular in human cells. By simultaneously studying *WDR62*-mutant cerebral organoids and KO mice, our studies provided novel insights into this issue. First, the proliferation of oRGs was reduced upon *WDR62* deletion. The oRGs are highly proliferative and their progenies contribute to the majority of upper layer neurogenesis[5,6,8]. Disruption of oRGs due to *WDR62* mutation may lead to the reduction of neurogenesis output specifically in the human cerebral cortex in comparison to that of mice, which could explain drastic organoid size reduction relative to the mild mouse microcephaly upon *WDR62* deletion. Secondly, *WDR62* mutant human NPCs and cerebral organoids exhibited robust ciliogenesis defects, including abnormally long cilia and an increased percentage of ciliated NPCs. In contrast, ciliogenesis defect was mild in mutant mice. These results suggest that the same cellular defects resulting from *WDR62* mutation may manifest more robustly in human NPCs than that in mice. Future studies should reveal whether and how those previously discovered functions of WDR62 in mice occur in human NPCs and cerebral organoids[12–14].

Previously, three *Wdr62* gene trap knockout (KO) mice have been reported, including one with an insertion between exons 14 and 15[12], one with an insertion between exons 21 and 22[14],

and one with an insertion in intron 21[13]. Here we generated and characterized a *Wdr62* complete KO mouse model. We found that centrosome proteins (WDR62-CEP170-KIF2A) are sequentially recruited to the basal body of the primary cilium, therefore ensuring appropriate cilium disassembly. Previously, we reported that *Wdr62* deletion disrupted mitotic progression coupled with cell death[12]. Using an independent gene trap line, Sgourdou et al. similarly discovered mitotic progression and cell death defect[14]. Although these mouse models are slightly different in their generation and focus on different aspects of Wdr62 biology, they all exhibit similar and subtle microcephaly phenotypes due to the NPC disruption. Furthermore, all of them displayed growth restrictions in addition to smaller brain sizes. It is known that the encephalization quotient is around 0.5 in mice, and is between 7.4 and 7.8 in humans[48]. The relative ratio of the cerebral cortex surface, which is associated with higher-order cognitive function, is 1:1000 between human and mouse[49]. Therefore, proportional dwarfism in the *Wdr62* KO mouse model may predict a selective brain size reduction in human patients.

The function and regulation of MCPH proteins in the primary cilium have seldom been studied despite the fact that centrosomes provide a template for ciliogenesis[25]. This WDR62–CEP170–KIF2A pathway represents the first example that microcephaly proteins act in a stepwise manner for their recruitments to the basal body to control ciliogenesis dynamics for NPC maintenance. Multiple evidences support this notion. First, cerebral organoids from independent *WDR62* mutant clones and a complete KO mouse model consistently showed ciliogenesis defect in NPCs. MEFs from our previously reported gene trap line also exhibited a cilium defect (data not shown). These results suggest that WDR62 regulates cilia across different species. Second, CEP170 and KIF2A have similar functions to that of WDR62 in regulating cilium disassembly, and they are sequentially recruited by WDR62 to the basal body of the primary cilium. Third, KIF2A overexpression is sufficient to rescue the deficits in cilium length, proliferation, and differentiation of *WDR62* mutant NPCs. It is likely that CEP170 and KIF2A are not the only factors that mediate WDR62 function in ciliogenesis. In addition to cilium regulation, WDR62 has been reported to have multiple functions, including mitotic progression[12,14], centriole biogenesis[13], centrosome asymmetry[31]. Future studies should determine to what extent disruption of individual WDR62 functions contribute to microcephaly.

Mutations in *KIF2A* cause microcephaly[46]. KIF2A promotes cilium disassembly and ensures proper cell cycle progression[26,47]. Unlike conventional kinesin motor proteins, KIF2A does not "walk" along microtubules but instead possesses microtubule-depolymerizing activity[50]. It has been reported that KIF2A drives cilium disassembly in a manner that is dependent on microtubule-depolymerizing activity[26]. Therefore, it is possible that WDR62 promotes cilium disassembly by enhancing microtubule-depolymerizing activity at the basal body of the primary cilium via KIF2A recruitment, disruption of which leads to abnormally long cilia. How does a cilium disassembly defect affect NPCs and thereby contribute to MCPH? Ciliogenesis and cell cycle progression are mutually exclusive[22,38]. Our data support the notion that a retardation of cilium disassembly would delay cell cycle re-entry and prolong the cell cycle length[23,25]. The cilium and microcephaly phenotypes arising from *WDR62* mutations resemble those from mutations in other genes associated with MCPH and Seckel syndrome[23,25]. Therefore, it is likely that cilium disruption coupled with NPC depletion observed in our *WDR62* mutants serve as a universal pathological mechanism underlying other genetic microcephaly.

## Methods

**Cell culture**. The human iPSC line (clone inventory code: R138363028) was characterized and obtained from NINDS Human Cell and Data Repository (NHCDR). H9 human ES cells were ordered from the Wisconsin International Stem Cell (WISC) Bank. ES cells or iPSCs (referred to here collectively as human pluripotent stem cells, hPSCs) were cultured in mTeSR-E8 medium (Stem cell technologies), and were passaged every week onto a new plate coated with Geltrex (Thermo Fisher Scientific). The hPSCs were detached from the plate by incubation with Accutase (Millipore) for 2 min. Individual colonies were then dissociated into small cell aggregates by manual pipetting. The medium was changed every day. MEFs were isolated from E14.5 embryos and cultured in Dulbecco's Modified Eagle's Medium (DMEM) with 15% fetal bovine serum, 1% penicillin/streptomycin and 1% Corning® glutagro™ Supplement. 293T cells were obtained from the American Type Culture Collection (ATCC) and cultured in DMEM (Invitrogen) supplemented with 10% FBS. All cell culture were handled according to protocols approved by the University of Southern California.

**CRISPR/Cas9 gRNA**. For genome targeting, gRNAs were designed using the online software Optimized CRISPR Design—MIT (http://crispr.mit.edu/) and cloned into pSpCas9(BB)-2A-Puro (PX459) V2.0 vector. PX459 V2.0 was a gift from Dr. Feng Zhang's laboratory (Addgene plasmid #62988). To evaluate the efficiency of targeting the human *WDR62* gene, the vectors were transiently transfected into HeLa cells. At 24 h after transfection, puromycin (4 μg/ml) was added into the medium for selection for 2 days. Genomic DNA of HeLa cells was collected by Phenol–Chloroform extraction. Genomic targeting regions were amplified using Phusion High-Fidelity PCR Master Mix (NEB). The primers were Wdr62-exon1-F:CTTCGCCCCCATTGGTTCTA; Wdr62-exon1-R: CATGGCAG GGAACAACCTGA;Wdr62-exon11-F:CTTCTGAGCTGCTCGTGCT and Wdr62-exon11-R: CCCTTCTGGAAGCCTCTTACG. The PCR products were denatured then annealed, and finally digested with T7 Endonuclease I (NEB) at 37 ℃ for 30 min. Genome-targeting efficiency was calculated as the ratio of intensity of digested fragments to total intensity of DNA fragments. The gRNAs with high-targeting efficiency, Exon1-gRNA (CTATGCGCGGAACGATGCAG) and Exon 11-gRNA (TTGGCTTCAGGCGACCGAAG), were chosen for the generation of *WDR62* knockout hPSCs.

**Generation of *WDR62* knockout hPSC lines**. The hPSCs were cultured with standard culture media with ROCK inhibitor Y27632 (10 μM, Selleckchem) for 24 h prior to electroporation. The cells were dissociated into single cells using Accutase followed by electroporation using Amaxa Nucleofector II eletroporator (Lonza) with Human Stem Cell Nucleofector Kit 1 (Lonza). In each electroporation, 3 μg plasmids were electroporated into $8.0 \times 10^5$ cells, which were then immediately plated on Geltrax-coated plates and cultured in mTeSR-E8 medium containing Y27632 (50 μM) for the first 24 h. Next, puromycin (0.5 μg/ml) was added to the medium for 2 days and then removed. Subsequently, hPSCs were maintained in medium without puromycin until colonies emerged. Individual colonies were picked up and expanded for 2 weeks. PCR products were generated and subjected to DNA sequencing to identify mutant clones.

**NPC induction and maintenance**. The hPSCs were cultured until 80% confluence in one well of a six-well plate, and then were passaged into one well of a 12-well plate until 100% confluent the next day. These cells were maintained for 2 weeks in neural induction medium, which consisted of N2B27 medium supplemented with dual Smad inhibitors SB431542 (10 μM) and LDN-193189 (0.1 μM) (Selleckchem). N2B27 medium was 50% DMEM/F12, 50% Neurobasal medium, 0.5% N2 supplement, 1% B27 supplement, 1% Glutamax, 1% penicillin–streptomycin, and 1% non-essential amino acids (Thermo Fisher Scientific). After 2 weeks, the cells were passaged as cell aggregates and underwent suspended culture for 2 days to form embryoid bodies (EBs). Then EBs were attached to the Geltrex-coated plates and cultured in neural induction medium containing 20 ng/ml bFGF until the neural rosettes emerged. Neural rosettes were manually picked up and dissociated into individual cells, which were then plated on Geltrex-coated plates. The human NPCs were maintained in N2B27 medium with 20 ng/ml bFGF and 20 ng/ml EGF, and the culture medium was changed every 2 days.

**Generation of cerebral organoids from hPSCs**. The hPSC colonies were dissociated into single cells using Accutase. On day 1, a total of 9000 cells were plated into each well of an ultra-low-attachment 96-well plate (Thermo Fisher Scientific) for single EB formation. The EB formation medium consisted of DMEM/F12, 20% Knockout Serum Replacement, 1% GlutaMAX, 1% non-essential amino acids, 50 μM ROCK Inhibitor Y27632, and 4 ng/ml bFGF (Perpotech). On day 4, the EBs were cultured in EB formation medium without Y27632 and bFGF. On day 7, the EBs were transferred to a low attachment 24-well plate (Corning) and cultured in neural induction medium for 5 days. The neural induction medium consisted of DMEM/F12, 0.5% N2 supplement, 1% GlutaMAX, 1% non-essential amino acids, 1% penicillin–streptomycin, and 10 μg/ml heparin with dual Smad inhibitors A83-01(1 μM) and LDN-193189 (0.1 μM). On day 12, the EBs were embedded into Matrigel droplets (Corning) and cultured for 4 days in medium containing 50% DMEM/F12, 50% Neurobasal medium, 0.5% N2 supplement, 1% B27 supplement without Vitamin A, 1% GlutaMAX, 1% non-essential amino acids, 1% penicillin–streptomycin, and 2.5 ng/ml human insulin. On day 16, the organoids were transferred to a spinner flask (WEATON) rotating continuously at 60 rpm and cultured in N2B27 medium with 2.5 ng/ml human insulin. The medium was changed every week. For long-term culture (more than 2 months) cerebral organoid, 5000 cells were used to generate EBs. After neural induction and Matrigel embedding, the cerebral organoids were transferred to the spinner flask and cultured in N2B27 medium with human insulin. On day 30, brain-derived neurotrophic factor (BDNF) was added to the flask at a concentration of 10 ng/ml. Medium was changed every 6 days.

**Immunostaining and immunohistochemical staining**. For human NPC immunostaining, cells were fixed in cold methanol for 8 min, washed twice with PBS, and then incubated with primary antibodies in blocking buffer (2% goat serum + 1% BSA + 0.1% TritonX-100 in PBS) overnight at 4 °C before secondary antibody incubation. Brain organoids were fixed in 4% PFA for 30 min at room temperature. Organoids were washed three times with PBS, incubated in 30% Sucrose solution at 4 °C overnight, and then embedded in OTC followed by dry ice freezing. The frozen organoids were sectioned into 10 μm-thick slices for immunohistochemical staining. For Sox2+/Pax6+ ventricular zone (VZ) thickness, measurements were taken at 45° angles to obtain the mean value (Supplementary Fig. 2B). To assess neuronal differentiation, we measured the relative thickness of neural layer (total layer thickness−VZ layer thickness)/total layer thickness (Supplementary Fig. 2C). Histological processing, TUNEL assay, and immunohistochemical labeling of cryosections were performed using cerebral cortex sections from different stages of embryos. The primary antibodies are listed in Supplementary Table 1. The secondary antibodies used were Alexa 488 and Alexa 555 conjugated to specific IgG types (Invitrogen Molecular Probes). All the experiments were repeated at least three times, and representative images are shown in the individual figures.

***Wdr62* knockout mouse generation**. The *Wdr62* targeting vector was generated through BAC recombineering services at University of North Carolina at Chapel Hill (UNC-CH). *Wdr62* exon 2 was replaced via homologous recombination by a loxP-exon2-frt-neo-frt-loxp targeting vector with 5′ and 3′ arms of homology. The Mouse Genetic Core Facility at National Jewish Health (NJH) (Denver, CO, USA) performed ES cell injection into C57BL/6N blastocysts. The chimeric offspring were mated to 129S1/SvImJ mice for germline transmission; the germline-transmitted heterozygous females were crossed with CMV-Cre males to remove exon 2, which yields a premature stop codon and ablates the Wdr62 protein. The PCR primers used for genotyping were: Wdr62 primer C: GGAACTTCAGT GCTTTGGTTTGC; Wdr62 primer I: GAAGAGGCTCCTTTCACTGCCT. All animals were handled according to protocols approved by the Institutional Animal Care and Use Committee at the University of Southern California.

**Cell cycle kinetics calculation**. Cell cycle kinetics were determined by a dual thymidine marker approach. On E16.5, the pregnant mouse was injected with CldU (10 mg/ml, 100 μl per 100 g body weight) at a time designed as $T = 0$ h, such that all cells at S-phase from the beginning of the experiment were labeled with CldU. At $T = 1.5$ h, the pregnant mouse was injected with EdU (1 mg/ml, 100 μl per 100 g body weight) to label all cells in S-phase at the end of the experiment. The animal was euthanized at $T = 2$ h and embryos were collected immediately. Embryo sections were immunostained using anti-BrdU antibody (Abcam, ab6326) and Click-iT® EdU Alexa Fluro® 555 Imaging Kit (Life Technologies, C10338). Images were obtained on a Zeiss LSM 710 inverted confocal microscope. The length of S-phase ($T_s$) and total length of cell cycle ($T_c$) were determined based on the relative number of cells that incorporated CldU and/or EdU. The ratio of the length of any one period of the cell cycle to that of another period is equal to the ratio of the number of cells in the first period to the number in the second period. Thus, the length of S phase ($T_s$) was calculated as the interval between the two injections ($T_i = 1.5$ h) divided by the quotient of the density of CldU+EdU− cells (cells that were in S phase but left it before EdU injection) and CldU+EdU+ cells (cells remaining in S phase at the end of the experiment). The equation used was $T_s = 1.5$ h$/(N_{CldU+EdU−}/N_{CldU+EdU+})$. The total length of the cell cycle was determined by the quotient of the density of CldU + EdU + and total proliferating cells using $T_c = T_s/(N_{CldU+EdU+}/N_{total})$.

**Purification of Flag-Wdr62 protein**. Expression constructs for Flag-Wdr62 were transfected into 293T cells followed by selection on puromycin (5 μg/ml) for stable cell line generation. Crude lysate was collected in lysis buffer (20 mM Tris–HCl [pH 7.5], 137 mM NaCl, 1 mM EDTA, 1.5 mM MgCl₂, 10% glycerol, 0.2% Triton X-100). To purify the Flag-Wdr62 proteins, cytoplasmic extract (S100) was incubated with anti-Flag M2 affinity gel (Sigma). Two washes were performed using buffer A [20 mM Tris–HCl (pH 7.9), 0.5 M KCl, 10% glycerol, 1 mM EDTA, 5 mM dithiothreitol (DTT), 0.2 mM phenylmethylsulfonyl fluoride (PMSF), 0.5% NP-40] followed by one wash with buffer B [20 mM Tris–HCl (pH 7.9), 0.1 M KCl, 10% glycerol, 1 mM EDTA, 5 mM DTT, 0.2 mM PMSF]. The affinity column was then eluted with 400 μg/ml Flag peptide. Proteins were further separated with Superose 6 gel filtration chromatography. Fractions were examined by SDS–PAGE analysis and silver staining, and were subject to mass spectrometry analyses.

**Western blot analysis**. Protein lysates were prepared with radio immunoprecipitation assay (RIPA) buffer. Protein lysates were separated utilizing sodium dodecyl sulfate polyacrylamide gel electrophoresis (SDS–PAGE), and were incubated with antibodies listed in Supplementary Table 1 followed by horseradish peroxidase (HRP)-labeled secondary antibodies (Bio-Rad, USA). For individual studies, the densitometry of individual blot signals from three independent western blot experiments were quantified using Image J software. The individual values for each blot signal was normalized to respective controls followed by the statistical analysis among different samples (Student's *t*-test).

**Immunoprecipitation**. Constructs containing flag-tagged full length, N-terminal, or C-terminal Wdr62 were transfected into 293T cells and cultured for 48 h before lysate collection. Cells were lysed in lysis buffer (50 mM Tris–HCl [pH7.4], 150 mM NaCl, 1 mM EDTA, 1% Triton X-100, and 1 tablet protease inhibitor [Roche] per 10 ml). Cell debris was pelleted at 13,000 r.p.m. for 15 min at 4 °C. For immunoprecipitation, 1–2 mg supernatant from the centrifugation of the lysate was incubated with anti-Flag M2 affinity resin (Sigma) at 4 °C for 3 h followed by anti-Flag beads for additional 2 h. The resin was washed with lysis buffer four times, and the bound proteins were analyzed by SDS–PAGE and immunoblotting. Twenty micrograms of whole lysate was loaded into the input lane. Western blots were performed using the antibodies described in Table S1.

**Cep170 knockdown lentivirus production**. Two Cep170 shRNA plasmids were purchased from Sigma-Aldrich:

TRCN0000243677: CCGGGATTAGACAATCCATTGATAACTCGAGTTATC
AATGGATTGTCTAATCTTTTTG

TRCN0000243675: CCGGGGCGCTTTCCTACTGATTATGCTCGAGCATAA
TCAGTAGGAAAGCGCCTTTTTG (targeting the same sequence in mouse and human). To produce lentivirus, a total of 15 µg knockdown plasmids were transfected into 293T cell together with 6 µg envelope plasmid VSVG and 9 µg packaging plasmid pspax2. The viruses in the supernatant were collected at 24 and 48 h after transfection, cleared by low speed centrifugation, and filtered through 0.22 µm pore size cellulose acetate filters. To knock down Cep170 in MEFs, cells were cultured in DMEM with 15% FBS plus 1% penicillin/streptomycin and Glutagro™ Supplement. At around 60% confluency, cells were infected by lentivirus expressing Cep170 shRNA for 8–12 h, followed by puromycin (2 µg/ml) selection for 72 h before experiments. To knock down CEP170 in human NPCs, the cells were infected by lentivirus together with polybrene (4 µg/ml) for 12 h. Puromycin (1 µg/ml) was added to the medium at 48 h after infection. Human NPCs were under puromycin selection for 2 days before experiments.

**Kif2a overexpression vector construction**. The pEGFP-Kif2A was a gift from Gohta Goshima & Ryota Uehara (Addgene plasmid # 52401). The fragment of EGFP fused with Kif2A was collected by digesting pEGFP-Kif2A with BamHI and AgeI, then ligated with lentivirus vector. The lentivirus vector backbone was from lentiCRISPRv2 puro vector, which was removed from the fragment of Cas9. The lentiCRISPRv2 puro vector was a gift from Brett Stringer (Addgene plasmid # 98290). The procedure of EGFP-Kif2A lentivirus production and infection of NPCs proceeded as described above in the section "Cep170 knockdown lentivirus production".

**Cilium biogenesis and cell cycle re-entry analyses**. To induce cilium assembly, MEFs were cultured on plates to reach >90% confluency. The culture medium was then replaced by starvation medium (DMEM with 0.5% FBS, 1% penicillin/streptomycin, and 1% Corning® Glutagro™ supplement) for 24 h before experiments. To promote cilium disassembly, MEFs under starvation were re-exposed to normal MEF culture media (DMEM with 15% fetal bovine serum, 1% penicillin/streptomycin, and 1% Corning® Glutagro™ supplement) for various periods of time before experiments. MEFs were fixed with pre-chilled 100% methanol for 10 min at −20 °C, and stained with different antibodies as listed in the figures and figure legends. For cell cycle re-entry assay, MEFs were serum-starved for 24 h followed by growth stimulation. DNA-replicating cells were pulse-labeled with 100 µg/ml ehynyl-deoxyuridine(EdU) for 3 h before fixation. EdU staining was performed as described in the Click-iT EdU Imaging Kit (C10338, Invitrogen).

**KIA2A rescued brain organoid generation**. Briefly, the *WDR62*⁻/⁻ iPSCs were dissociated into single cells by Accutase, then were plated on Geltrex-coated dish. The medium contained ROCK inhibitor Y-27632 (10 µM), KIF2A-EGFP lentivirus particles and polybrene (8 µg/ml). The medium was changed after 12 h infection. After one week, the EGFP-positive iPS colonies were picked and expanded. The KIF2A-EGFP-positive iPSCs were used to generate KIF2A rescue brain organoid following the protocol described previously.

**In utero electroporation**. DNA constructs including pCAG-H2BGFP and Cep170 shRNA plus 0.5% Fast Green (Sigma) were injected into the lateral ventricle of E14.5 wild-type embryos followed by electroporation using an ECM 830 electroporator (BTX) with four 100 ms pulses separated by 100 ms intervals at 30 V. To improve retention of the pregnancy, we avoided plasmid injection and electroporation of the two embryos next to the ovaries and the two embryos next to the upper vagina. Cep170 shRNAs were purchased from Sigma. In utero development was allowed to continue for 48 h (or as indicated), then the embryos were dissected and the cortical region was processed for immunohistochemistry analyses.

**Reporting summary**. Further information on research design is available in the Nature Research Reporting Summary linked to this article.

## Data availability

All relevant data are available from the corresponding authors upon reasonable request.

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

## Acknowledgements

We thank Chen laboratory colleagues for stimulating discussions. We are grateful for Bridget Samuels's critical reading of the manuscript. We are grateful for the Cep164 antibody from Dr. Erich A. Nigg's lab, Ift88 antibody from Dr. Bradley K. Yoder's lab, and Kif24 antibody from Dr. Brian David Dynlacht's lab. Chen laboratory is supported by funds from the Associate Dean of Research Fund from the Center for Craniofacial Molecular Biology, Herman Ostrow School of Dentistry at the University of Southern California, and grants R01NS097231 (J.C.) and R01NS096176 (J.C.) from the National Institute of Health (NIH).

## Author contributions

W.Z., S.-L.Y., M.Y., S.H., E.F., Q.S., J.L.C. conceived and performed all experiments. B.E. H., Y.-H.S., H.L., M.-L.G., S.B., A.-M.L., Z.Z., J.X., Z.-P.L. helped with the manuscript writing. J.-F.C. designed and interpreted the experiments and wrote the manuscript. The data supporting the findings of this study are available from the corresponding author upon reasonable request.

## Additional information

**Competing interests:** The authors declare no competing interests.

**Journal Peer Review Information:** *Nature Communications* thanks the anonymous reviewer(s) for their contribution to the peer review of this work. Peer reviewer reports are available.

