## [Peer Review File · Nature Communications]

Reviewers' comments:

Reviewer #1 (Remarks to the Author):

The manuscript by Zhang et al. studies the role of WDR62 - a centrosomal protein - in the etiology of microcephaly. By creating WDR62 knockout pluripotent stem cells (PSC) and thereof derived neural progenitor cells (NPC) and cerebral organoids as well as WDR62 mutant mouse models they found that WDR62 ablation led to retarded cilium disassembly, increase in cilium length and delayed cell cycle progression resulting in decreased proliferation and early neurogenesis. On a molecular level they found that WDR62 interacts with CEP170 impacting its localization to the basal body of primary cilium and its recruitment of the microtubule-depolymerizing factor KIF2A to disassemble cilium. By KIF2A overexpression they could partially rescue WDR62 induced phenotypic changes in NPC and cerebral organoids.

The manuscript is well written and shows a comprehensive analysis of the role of WDR62 on cilium number, length and disassembly. By using different model systems ranging from mouse in vivo to human in vitro systems the authors tried to elucidate human specific aspects of WDR62 induced microcephaly. Here they could convincingly show a role of the WDR62-CEP170-KIF2A protein pathway in the increased expansion of neuroepithelial cells especially in the human system. However, the analyze on outer radial glial cells, a progenitor population which is thought to contribute to the increased size and complexity of the human brain, falls short.

Specific points

1. In the first part the authors describe their PSC-derived WDR62 knockout lines and thereof derived cerebral organoids. They performed a comprehensive analyze of the size of the WDR62-KO organoids and isogenic controls. They then investigated the overall structure of the organoids and quantified oRGC proliferation. In general the authors could think of showing the data of the overall size in a more condensed form by including one example of a WDR62-KO and isogenic control derived organoid in the main figure and moving the other images to the supplementary information. The size development of the organoids over time could also be combined in one graph.
2. The analyses of the VZ-thickness gives a nice overview on the overall structural changes due to WDR62 depletion however the representative images bring up the question whether the morphology and organization of the cells in the VZ differ in WDR62-KO derived organoids. They seem to be smaller than in the control and less organized. The authors should either exchange the images or comment on these phenomena. In addition, structural organization in the given Sox2 / PTPRZ1 images are not convincing. From these images it is also not clear how they distinguish between VZ and SVZ. It would be better to include overview images of the Sox2 / PTPRZ1 stained organoids instead of cutting out an area and indication VZ and SVZ areas in the images.
2. In mice the authors observed increased apoptosis in the WDR62 mutant. This was also a phenotypic change already described in the literature. It would be interesting to show whether this is also true for the WDR62 KO derived organoids.
3. It is known that FGF/EGF expanded NPC adapt over the passages a posterior phenotype The authors should characterize the regional identity of the generated NPC and indicate the passage numbers of their characterization and the passage numbers when the cells were used for analyses. This would give an overview whether the applied NPC represent a dorsal cortical identity.
4. When it comes to the analyses of the cell cycle the authors show quite convincing data on cell

proliferation, however, the analyses of premature differentiation could be underlined with additional data sets. First, the Nestin / BIII-Tubulin images (Figure 6 H) of the NPCs in 2D do not reflect the quantitative results (Figure 6 I). Second, Figure 6 J shows Sox2 positive cells above the indicated CP region in control cultures. Overview images should be shown to understand why and how the authors define the different areas analyzed. Third, it would be interesting to analyze the plane of cell division of apical RGCs in the organoids. It was described that mutations or KO of WDR62 leads to a switch from symmetric to asymmetric cell division. Can that be recapitulated in the WDR62-KO organoids. And does the rescue with KIF2A has an impact on the division plane of the apical RGCs?

5. The authors analyze the effect of KIF2A overexpression on NPCs in 2D and overall organoid size. Here the analyses of the proliferation rate of oRGCs in the WDR62-KO-Kif2A overexpressing organoids would be interesting. This could contribute to the understanding why mice show reduced phenotypic changes compared to humans.

Minor points:

The authors should change the wording when describing the general structural organization of the organoids. They write: ...“suggesting that the general layer organization was relatively preserved”...as they did not analyze cortical layering by using different cortical layer specific markers but rather analyzed the overall structure (VZ, SVZ and CP) they should adapt the wording respectively.

Reviewer #2 (Remarks to the Author):

Zhang et al. present comprehensive functional studies of the microcephaly-causing gene, WDR62, based on null mutant and human PSC-carrying WDR 62 deletion mutations generated through CRISPR-Cas9 technology. While previous studies have focused on mitotic delay and apoptotic cell death in conjunction with centrosome function, in this manuscript the authors concentrate on abnormal overgrowth of cilia and protraction of the cell cycle and the contribution of these perturbations to microcephaly pathology. The authors claim that WDR62 plays a role in disassembling cilia that is important for normal cell cycle progression and that cell cycle delay causes precocious neuronal differentiation, which reduces the number of cycling cells and generates microcephaly. To provide a molecular mechanism, they identified centrosomal protein Cep170 as an interacting protein of WDR62, which recruits Kif2A, a kinesin with microtubule depolymerizing activity, at the base of cilia. The overexpression of Kif2A using lentivirus can restore the proliferating progenitor population and the normal length of NPC cilia and partial recovery of the surface area of WDR-deficient human cortical organoids. Overall, the data are convincing and the use of multiple models, including in vivo (mouse), ex vivo (human organoids) and in vitro (human NPC, mouse MEF), is also a strength. While the study has the potential to provide a new pathogenic mechanism for microcephaly caused by WDR62 mutations and new insight into the role of the Wdr62-Cep17-Kif2A module in cilia disassembly and cell cycle regulation, the following issues should be addressed to draw firm and meaningful conclusions.

Major points

(1) It is unclear that the mild cilia defects found in Wdr62^{-/-} mouse mutant are responsible for the increased cell cycle length and the reduction of progenitors and cortical thickness. Is there any evidence for delayed disassembly of cilia in mouse NPCs? For example, do mutants have a greater number of cilia in the ventricular surface despite a reduced number of progenitors?

(2) Thorough cell cycle length analyses should be performed to establish that the length of G1 is extended in Wdr62 mutants. Are there any changes in the length of S-phase? S- phase length should be compared in addition to total cell cycle length. Furthermore, because a previous study showed that mitosis is significantly delayed in the gene trap alleles of Wdr62, the contribution of changes in the length of mitosis to total cell cycle length should also be addressed.

(3) The precocious neuronal production in WDR62^{-/-} organoids is demonstrated by the expansion of the DCX-expressing domain in human brain organoids. However, CTIP2 positive neurons are reduced in number. Have the authors detected any cell death and/or increased mitotic cells in organoids? It is possible that delayed mitosis and apoptotic cell death of progenitors are the main causes of the reduced organoid size and the decreased number of progenitors and postmitotic neurons.

(4) Furthermore, precocious generation of neurons is not explored in the mouse model. Since cell cycle length is increased yet progenitor population is reduced, compromised cell survival or precocious generation of neurons could be responsible for reduced progenitor pools. Is there any evidence for precocious generation of neurons in the mouse mutants? To provide evidence that a lengthened cell cycle causes precocious differentiation, at least in mice, the authors could carry out a fraction study. For example, they could show an increase in the number of cells quitting the cell cycle by demonstrating a reduction of BrdU positive cells overlapping with a progenitor marker 24 hrs after BrdU injection.

(5) Kif2A is also expressed in the spindle pole and is known to have a centrosomal function in normal mitosis. It is important to determine whether Kif2A overexpression has effects on the mitotic cells, such as reducing the number of cells in mitosis.

(6) Fig 7 E, F, G, H, I J and Fig S8 A, B, C, F, G, H are identical and figure legend for S8 is missing.

Minor points

Human control should not be referred to as WT. This confuses the animal experiments with the human brain organoids and the terminology is inadequate, at least to this reviewer.

Figure 3, L does not correspond to the boxed area of K. it appears to be a view of the ventricular surface rather than a cross-sectional view of cortex.

Point-by point responses to the reviewers' comments

Dear Editors and Reviewers,

We appreciate the careful reviews and comments that have helped to improve the quality of the manuscript. We agree with all of the criticisms and therefore we have performed substantial new experiments to address these concerns. We fully addressed reviewers' overlapping concerns by experiments, including the validation of apoptosis in brain organoids (Fig. 2I-2L), precocious differentiation of NPCs (Fig. 6L-6O), mitotic defects (Fig. 2E-2F), and rescue effects of Kif2A overexpression in mitosis in organoids (Fig. 8L). In addition, we have performed additional experiments and addressed other concerns from individual reviewers, including symmetric/asymmetric division (Fig. 2G-2H), cilium numbers in mutant mice (Fig. 4J), cell cycle analysis (Fig. J-4K). Below we detail our responses to the reviewers' comments.

Reviewer #1:

The manuscript by Zhang et al. studies the role of WDR62 - a centrosomal protein - in the etiology of microcephaly. By creating WDR62 knockout pluripotent stem cells (PSC) and thereof derived neural progenitor cells (NPC) and cerebral organoids as well as WDR62 mutant mouse models they found that WDR62 ablation led to retarded cilium disassembly, increase in cilium length and delayed cell cycle progression resulting in decreased proliferation and early neurogenesis. On a molecular level they found that WDR62 interacts with CEP170 impacting its localization to the basal body of primary cilium and its recruitment of the microtubule-depolymerizing factor KIF2A to disassemble cilium. By KIF2A overexpression they could partially rescue WDR62 induced phenotypic changes in NPC and cerebral organoids.

The manuscript is well written and shows a comprehensive analysis of the role of WDR62 on cilium number, length and disassembly. By using different model systems ranging from mouse in vivo to human in vitro systems the authors tried to elucidate human specific aspects of WDR62 induced microcephaly. Here they could convincingly show a role of the WDR62-CEP170-KIF2A protein pathway in the increased expansion of neuroepithelial cells especially in the human system. However, the analyze on outer radial glial cells, a progenitor population which is thought to contribute to the increased size and complexity of the human brain, falls short.

Specific points

1. In the first part the authors describe their PSC-derived WDR62 knockout lines and thereof derived cerebral organoids. They performed a comprehensive analyze of the size of the WDR62-KO organoids and isogenic controls. They then investigated the overall structure of the organoids and quantified oRGC proliferation. In general the authors could think of showing the data of the overall size in a more condensed form by including one example of a WDR62-KO and isogenic control derived organoid in the main figure and moving the other images to the supplementary information. The size development of the organoids over time could also be combined in one graph.

Response: Done (Fig. 1D-1E, Fig. S3A). Thank you for the suggestions.

2. The analyses of the VZ-thickness gives a nice overview on the overall structural changes due to WDR62 depletion however the representative images bring up the question whether the morphology and organization of the cells in the VZ differ in WDR62-KO derived organoids. They seem to be smaller then in the control and less organized. The authors should either exchange the images or comment on these phenomena. In addition, structural organization in the given Sox2 / PTPRZ1 images are not convincing. From these images it is also not

clear how they distinguish between VZ and SVZ. It would be better to include overview images of the Sox2 / PTPRZ1 stained organoids instead of cutting out an area and indication VZ and SVZ areas in the images.

Response: Agreed. We did not find repeatable and robust changes in the morphology and organization of VZ cells, and have replaced the previous VZ thickness images (Fig. 2A-2B). We also include the overview of Sox2/PTPRZ1 stained organoids (Fig. 2C-2D).

2. In mice the authors observed increased apoptosis in the WDR62 mutant. This was also a phenotypic change already described in the literature. It would be interesting to show whether this is also true for the WDR62 KO derived organoids.

Response: Yes. This increased apoptosis is also true in mutant organoids as evidenced by TUNEL and Caspase-3 staining (Fig. 2I-2L).

3. It is known that FGF/EGF expanded NPC adapt over the passages a posterior phenotype. The authors should characterize the regional identity of the generated NPC and indicate the passage numbers of their characterization and the passage numbers when the cells were used for analyses. This would give an overview whether the applied NPC represent a dorsal cortical identity.

Response: Thank you for pointing it out! Passage number is within 3 after NPC generation. Pax6-positive cells is more than 90% (Fig. 6A-6B), suggesting the dorsal cortical identity.

4. When it comes to the analyses of the cell cycle the authors show quite convincing data on cell proliferation, however, the analyses of premature differentiation could be underlined with additional data sets. First, the Nestin / β III-Tubulin images (Figure 6 H) of the NPCs in 2D do not reflect the quantitative results (Figure 6 I). Second, Figure 6 J shows Sox2 positive cells above the indicated CP region in control cultures. Overview images should be shown to understand why and how the authors define the different areas analyzed. Third, it would be interesting to analyze the plane of cell division of apical RGCs in the organoids. It was described that mutations or KO of WDR62 leads to a switch from symmetric to asymmetric cell division. Can that be recapitulated in the WDR62-KO organoids. And does the rescue with KIF2A has an impact on the division plane of the apical RGCs?

Response: Thank you for the valuable suggestions! We have performed additional experiments according to your suggestions, including the overview SOX2/DCX IHC in brain organoids (Fig. 6M-6O), and Western blot analyses of differentiation markers TuJ1 and Neurofilament (Fig. 6L). All these results are consistent with a precocious differentiation phenotype in mutant NPCs.

To study symmetric/asymmetric division, we have combined p-VIM and Hoechst staining to define cell division angles. Quantification results showed a decreased horizontal division and an increased vertical division (Fig. 2G-2H).

5. The authors analyze the effect of KIF2A overexpression on NPCs in 2D and overall organoid size. Here the analyses of the proliferation rate of oRGCs in the WDR62-KO-Kif2A overexpressing organoids would be interesting. This could contribute to the understanding why mice show reduced phenotypic changes compared to humans.

Response: Author agreed with you. Our KIF2A rescue was performed in 2D NPCs. We have tried similar studies in brain organoids for oRG cells. However, it is technically challenging and we cannot make convincing conclusions due to the big variations from statistical analyses.

Minor points:

The authors should change the wording when describing the general structural organization of the organoids. They write: ... "suggesting that the general layer organization was relatively preserved" ...as they did not

analyze cortical layering by using different cortical layer specific markers but rather analyzed the overall structure (VZ, SVZ and CP) they should adapt the wording respectively.

Response: Corrected. Please see revised text. Thank you for pointing it out!

Reviewer #2:

Zhang et al. present comprehensive functional studies of the microcephaly-causing gene, WDR62, based on null mutant and human PSC-carrying WDR 62 deletion mutations generated through CRISPR-Cas9 technology. While previous studies have focused on mitotic delay and apoptotic cell death in conjunction with centrosome function, in this manuscript the authors concentrate on abnormal overgrowth of cilia and protraction of the cell cycle and the contribution of these perturbations to microcephaly pathology. The authors claim that WDR62 plays a role in disassembling cilia that is important for normal cell cycle progression and that cell cycle delay causes precocious neuronal differentiation, which reduces the number of cycling cells and generates microcephaly. To provide a molecular mechanism, they identified centrosomal protein Cep170 as an interacting protein of WDR62, which recruits Kif2A, a kinesin with microtubule depolymerizing activity, at the base of cilia. The overexpression of Kif2A using lentivirus can restore the proliferating progenitor population and the normal length of NPC cilia and partial recovery of the surface area of WDR-deficient human cortical organoids. Overall, the data are convincing and the use of multiple models, including in vivo (mouse), ex vivo (human organoids) and in vitro (human NPC, mouse MEF), is also a strength. While the study has the potential to provide a new pathogenic mechanism for microcephaly caused by WDR62 mutations and new insight into the role of the Wdr62-Cep17-Kif2A module in cilia disassembly and cell cycle regulation, the following issues should be addressed to draw firm and meaningful conclusions.

Major points

(1) It is unclear that the mild cilia defects found in Wdr62^{-/-} mouse mutant are responsible for the increased cell cycle length and the reduction of progenitors and cortical thickness. Is there any evidence for delayed disassembly of cilia in mouse NPCs? For example, do mutants have a greater number of cilia in the ventricular surface despite a reduced number of progenitors?

Response: Yes. We found an increase in cilia numbers in mutant ventricular surface (Fig. 4J). Our data suggest that cilia defect may contribute to microcephaly, and likely is not the only cause for microcephaly.

(2) Thorough cell cycle length analyses should be performed to establish that the length of G1 is extended in Wdr62 mutants. Are there any changes in the length of S-phase? S-phase length should be compared in addition to total cell cycle length. Furthermore, because a previous study showed that mitosis is significantly delayed in the gene trap alleles of Wdr62, the contribution of changes in the length of mitosis to total cell cycle length should also be addressed.

Response: Yes. S-phase length is reduced in mutants (Fig. 5I). Our previous studies (Chen et al., Nat. Communi. 2014) showed that mitosis is impaired, which was confirmed in current new *Wdr62* null mouse models (Fig. S5C-5F) and brain organoids (Fig. 2E-2F, 2I-2L). In the future, we like to investigate the relative contributions of mitotic delay and cilia defect to the cell cycle length change.

(3) The precocious neuronal production in WDR62^{-/-} organoids is demonstrated by the expansion of the DCX-expressing domain in human brain organoids. However, CTIP2 positive neurons are reduced in number. Have the authors detected any cell death and/or increased mitotic cells in organoids? It is possible that delayed mitosis and apoptotic cell death of progenitors are the main causes of the reduced organoid size and the decreased number of progenitors and postmitotic neurons.

Response: Thank you for the valuable suggestions. We have performed experiments accordingly. We found an increase in mitotic cells (Fig. 2E-2F). There is an increased apoptotic cell death in mutant organoids (Fig. 2I-

2L). These data are consistent with our previous publication and support your idea that delayed mitosis and cell death of NPCs contribute to reduced organoid sizes. Meanwhile, our data suggest cilia defect coupled with cell cycle disruption also contributes to microcephaly. We are not clear that which one (mitosis vs. cilia defect) contribute more to microcephaly at this moment, which could be investigated in the future.

(4) Furthermore, precocious generation of neurons is not explored in the mouse model. Since cell cycle length is increased yet progenitor population is reduced, compromised cell survival or precocious generation of neurons could be responsible for reduced progenitor pools. Is there any evidence for precocious generation of neurons in the mouse mutants? To provide evidence that a lengthened cell cycle causes precocious differentiation, at least in mice, the authors could carry out a fraction study. For example, they could show an increase in the number of cells quitting the cell cycle by demonstrating a reduction of BrdU positive cells overlapping with a progenitor marker 24 hrs after BrdU injection.

Response: Authors agreed with you and have performed the experiment you suggested. A fraction study revealed a precocious exit of cell cycle by mutant neural progenitor cells (Fig. 5J-5K).

(5) Kif2A is also expressed in the spindle pole and is known to have a centrosomal function in normal mitosis. It is important to determine whether Kif2A overexpression has effects on the mitotic cells, such as reducing the number of cells in mitosis.

Response: Thank you for the suggestions. We have examined mitotic cells in KIF2A overexpressed organoids and found a reduced number of cells in mitosis (Fig. 8L).

(6) Fig 7 E, F, G, H, I J and Fig S8 A, B, C, F, G, H are identical and figure legend for S8 is missing.

Response: Authors apologized for the mistakes and have corrected them.

Minor points

Human control should not be referred to as WT. This confuses the animal experiments with the human brain organoids and the terminology is inadequate, at least to this reviewer.

Figure 3, L does not correspond to the boxed area of K. it appears to be a view of the ventricular surface rather than a cross-sectional view of cortex.

Response: Corrected. Thank you for pointing it out!

Reviewers' comments:

Reviewer #1 (Remarks to the Author):

The revised manuscript by Zhang et al include several new data sets / images / quantifications. The authors have performed additional experiments / analyses to address open questions. Some of the new data points open new question and some of the mentioned points were not addressed. In detail:

Point 1: Best would be to include one control image and one WDR62-KO image of each time point (4 weeks, 5 weeks and 6 weeks) in the main figure as representative and move the other lines into the supplementary information.

Point 2: Figure 2C: indicate in the figure legend the cell line the images were taken from (control? WDR62-KO?)

Point 2: Analyses of apoptosis: Figure 2 I-L: it is not clear how the authors distinguish between the SV/SVZ and the CP-like region. Co-stainings for apoptotic markers (Tunel or Caspase 3) with e.g. a pan neuronal marker such as β III-Tubulin would help to distinguish the CP like region from the SV/SVZ. One could also think of co-staining with a progenitor marker such as Sox2 and a neuronal marker such as β III Tubulin and quantify apoptosis in progenitors and neurons without specifying the zones. In addition, the caspase 3 AB is not specified nor included in Table 1. Only cleaved caspase 3 represents the active form and by that is an indication for apoptotic cells.

Point 4: The representative images for the cell cycle and precocious differentiation in 2D (Fig. 6J) still do not reflect the results of the corresponding quantification in Fig.6 K. The authors should include lower magnification overview images supporting their quantitative data. The results from the Western Blot was not quantified. The authors should discuss / explain why they did not detect any NF band but a β III-Tubulin band in the control cultures – which NF was used? The description of the Western blot is missing in the methods section, the NF antibody is not specified in Table 1.

Point 4/5: The authors addressed cell division of apical RGCs in the revised version of the manuscript in WT and WDR62-KO organoids confirming an increase in vertical/oblique dividing cells. They however did not address whether KIF2A rescue has an impact on the division plane.

Point 5: The authors show data on KIF2A rescue organoid (Fig8K). According to these data the KIF2A rescue organoids exhibit a significant increase in surface area and decrease in pVim positive cells in the VZ compared to the WDR2-KO derived organoids. They claim that analyses of oRGCs is not possible due to high variations in the system. The authors should include additional data such as the division plane of the p-Vim positive cells and CP thickness to further investigate the impact of the KIF2A rescue in organoids (similar as what has been analyzed in WT and WDR62-KO organoids). They should also include representative images of the immunostained rescue organoids. In addition, they should specify in the methods how they generate the KIF2A-rescue organoids.

Minor points:

Figure 2D: adjust position of the labeling WDR62-/- (this applies to multiple graphs in the different Figures)

The authors should always use β III Tubulin to specify Tuj stainings as Tuj is just the used clone. The antibody is not anti-Tuj but anti- β III Tubulin.

Reviewer #2 (Remarks to the Author):

Acceptable. All of the previous concerns were addressed.

Point-by point responses to the reviewers' comments

Dear Editors and Reviewers,

We appreciate the careful reviews and comments that have helped to improve the quality of the manuscript. We agree with all of the criticisms and therefore we have performed new studies, re-analyzed previous data, and revised manuscript to address these concerns. Below we detail our responses to the reviewers' comments.

Reviewer #1 (Remarks to the Author):

The revised manuscript by Zhang et al include several new data sets / images / quantifications. The authors have performed additional experiments / analyses to address open questions. Some of the new data points open new question and some of the mentioned points were not addressed. In detail:

Point 1: Best would be to include one control image and one WDR62-KO image of each time point (4 weeks, 5 weeks and 6 weeks) in the main figure as representative and move the other lines into the supplementary information.

Response: Agreed and done (**Fig. 1D**).

Point 2: Figure 2C: indicate in the figure legend the cell line the images were taken from (control? WDR62-KO?)

Response: It is the control. Thank you!

Point 3: Analyses of apoptosis: Figure 2 I-L: it is not clear how the authors distinguish between the SV/SVZ and the CP-like region. Co-stainings for apoptotic markers (Tunel or Caspase 3) with e.g. a pan neuronal marker such as β III-Tubulin would help to distinguish the CP like region from the SV/SVZ. One could also think of co-stainig with a progenitor marker such as Sox2 and a neuronal marker such as β III Tubulin and quantify apoptosis in progenitors and neurons without specifying the zones. In addition, the caspase 3 AB is not specified nor included in Table 1. Only cleaved caspase 3 represents the active form and by that is an indication for apoptotic cells.

Response: Authors agreed. We now included the SOX2 to label NPCs (**Fig. 2I-2L**). We have included active caspase-3 antibody information in Table S1.

Point 4: The representative images for the cell cycle and precocious differentiation in 2D (Fig. 6J) still do not reflect the results of the corresponding quantification in Fig.6 K. The authors should include lower magnification overview images supporting their quantitative data. The results from the Western Blot was not quantified. The authors should discuss / explain why they did not detect any NF band but a β III-Tubulin band in the control cultures – which NF was used? The description of the Western blot is missing in the methods section, the NF antibody is not specified in Table 1.

Response: Authors agreed. It is challenging to quantify the Nestin and TuJ1 based on immunostaining results since they both are expressed in the cytoplasm as you can tell from the images. Therefore, we have switched our 2D differentiation quantification to Western blot results.

We have quantified and described Western blot results and methods. The NF antibody reacts with both phosphorylated and non-phosphorylated forms of medium neurofilament protein (160 kDa) (Abcam, ab7794). NF is very low in WT NPCs cells as shown in the right figure with longer exposure.

Point 4/5: The authors addressed cell division of apical RGCs in the revised version of the manuscript in WT and WDR62-KO organoids confirming an increase in vertical/oblique dividing cells. They however did not address whether KIF2A rescue has an impact on the division plane.

Response: We have done the experiments and the result is negative. That is why we did not put it in the manuscript.

Point 5: The authors show data on KIF2A rescue organoid (Fig8K). According to these data the KIF2A rescue organoids exhibit a significant increase in surface area and decrease in pVim positive cells in the VZ compared to the WDR2-KO derived organoids. They claim that analyses of oRGCs is not possible due to high variations in the system. The authors should include additional data such as the division plane of the p-Vim positive cells and CP thickness to further investigate the impact of the KIF2A rescue in organoids (similar as what has been analyzed in WT and WDR62-KO organoids). They should also include representative images of the immunostained rescue organoids. In addition, they should specify in the methods how they generate the KIF2A-rescue organoids.

Response: Authors agreed. For KIF2A rescue study, we have added organoid imaging (**Fig. S9A**), VZ thickness as *WDR62^{-/-}* (**Fig. S9C**), and the representative imaging of p-H3 staining (**Fig. S9B**). Since division angle deficit was not rescued by KIF2A, we did not include that image. We have added the methods how the KIF2A-rescue organoids were generated.

Minor points:

Figure 2D: adjust position of the labeling WDR62^{-/-} (this applies to multiple graphs in the different Figures)

Response: Corrected. Thank you!

The authors should always use β III Tubulin to specify Tuj stainings as Tuj is just the used clone. The antibody is not anti-Tuj but anti- β III Tubulin.

Response: Agreed. We have changed it.

REVIEWERS' COMMENTS:

Reviewer #1 (Remarks to the Author):

The authors have addressed all the reviewer comments however there is one remaining open point which should be considered:

Connected to point 3: analyzes of apoptosis.

The Sox2/TUNEL and the Cleaved caspase 3 /Sox2 co-staining clearly helped to distinguish the VZ from the SVZ/CP. However the authors should also label with the indicated white line the border of the two analyzed sections. They state that "Above and below white dotted lines in K represent CP/SVA and VZ-like regions"

This border is clearly not on the same high in the control and the WDR62 patient as currently indicated in the figure. Indicate with the white line the boarder between the densely packed stratified Sox2 + cells and the more loosely arranged cells of the SVZ/CP and also quantify the apoptotic cells of the now clearly visible VZ and SVZ/CP, respectively.

Point-by point responses to the reviewers' comments

Dear Editors and Reviewers,

We appreciate the careful reviews and comments that have helped to improve the quality of the manuscript. Below we detail our responses to the reviewers' comments.

Reviewer #1 (Remarks to the Author):

The authors have addressed all the reviewer comments however there is one remaining open point which should be considered:

Connected to point 3: analyzes of apoptosis.

The Sox2/TUNEL and the Cleaved caspase 3 /Sox2 co-staining clearly helped to distinguish the VZ from the SVZ/CP. However the authors should also label with the indicated white line the border of the two analyzed sections. They state that "Above and below white dotted lines in K represent CP/SVA and VZ-like regions"

This border is clearly not on the same high in the control and the WDR62 patient as currently indicated in the figure. Indicate with the white line the boarder between the densely packed stratified Sox2 + cells and the more loosely arranged cells of the SVZ/CP and also quantify the apoptotic cells of the now clearly visible VZ and SVZ/CP, respectively.

Response: Agreed. We have changed it accordingly.